# Deletion of intestinal *Hdac3* remodels the lipidome of enterocytes and protects mice from diet-induced obesity

Mercedes Dávalos-Salas [1,2], Magdalene K. Montgomery[3], Camilla M. Reehorst [1,2], Rebecca Nightingale[1,2], Irvin Ng [1,2], Holly Anderton [1], Sheren Al-Obaidi[1], Analia Lesmana[1], Cameron M. Scott[1], Paul Ioannidis [1], Hina Kalra[4], Shivakumar Keerthikumar[4], Lars Tögel [1,2], Angela Rigopoulos[1,2], Sylvia J. Gong[1], David S. Williams[1,2,5], Prusoth Yoganantharaja[6], Kim Bell-Anderson [7], Suresh Mathivanan [4], Yann Gibert[6], Scott Hiebert[8], Andrew M. Scott[1,2,9], Matthew J. Watt[3]* & John M. Mariadason [1,2,9]*

Histone deacetylase 3 (*Hdac3*) regulates the expression of lipid metabolism genes in multiple tissues, however its role in regulating lipid metabolism in the intestinal epithelium is unknown. Here we demonstrate that intestine-specific deletion of *Hdac3* (*Hdac3^IKO*) protects mice from diet induced obesity. Intestinal epithelial cells (IECs) from *Hdac3^IKO* mice display co-ordinate induction of genes and proteins involved in mitochondrial and peroxisomal β-oxidation, have an increased rate of fatty acid oxidation, and undergo marked remodelling of their lipidome, particularly a reduction in long chain triglycerides. Many HDAC3-regulated fatty oxidation genes are transcriptional targets of the PPAR family of nuclear receptors, *Hdac3* deletion enhances their induction by PPAR-agonists, and pharmacological HDAC3 inhibition induces their expression in enterocytes. These findings establish a central role for HDAC3 in co-ordinating PPAR-regulated lipid oxidation in the intestinal epithelium, and identify intestinal HDAC3 as a potential therapeutic target for preventing obesity and related diseases.

[1] Olivia Newton John Cancer Research Institute, Melbourne, Victoria, Australia. [2] La Trobe University School of Cancer Medicine, Melbourne, Victoria, Australia. [3] Department of Physiology, Faculty of Medicine Dentistry and Health Sciences, University of Melbourne, Parkville, Victoria, Australia. [4] La Trobe Institute for Molecular Sciences, La Trobe University, Melbourne, Victoria, Australia. [5] Department of Pathology, Austin Health, Melbourne, Victoria, Australia. [6] Department of Medicine, Deakin University, Geelong, Victoria, Australia. [7] Faculty of Science, Charles Perkins Centre, University of Sydney, Sydney, NSW, Australia. [8] Vanderbilt University, Nashville, TN, USA. [9] Department of Medicine, University of Melbourne, Melbourne, Victoria, Australia. *email: matt.watt@unimelb.edu; john.mariadason@onjcri.org.au

Histone deacetylase 3 (HDAC3) is a class I HDAC which catalyses the removal of acetyl groups from lysine residues in histone and non-histone proteins[1]. By inducing the deacetylation of DNA-bound histones, HDAC3 acts primarily to repress transcription by promoting a closed chromatin conformation[1,2]. Comparatively, by catalysing the deacetylation of specific transcription factors, HDAC3 can increase or decrease their transcriptional activity depending on the molecular context[1,2].

HDAC3 is the catalytic component of the related nuclear receptor co-repressor (NCoR) and silencing-mediator of retinoic and thyroid receptors (SMRT) co-repressor complexes[3]. These large multi-subunit protein complexes are recruited by transcription factors to specific promoter regions to mediate transcriptional repression[4,5]. NCoR and SMRT serve as scaffolds and the principal point of contact between the co-repressor complex and its transcription factor partners, which include the retinoic acid receptor (RAR), retinoid X receptor (RXR) and peroxisome proliferator-activated receptor (PPAR) nuclear receptors[6,7].

HDAC3 performs several roles in intestinal and colonic epithelial cells. Alenghat et al. described a key role for Hdac3 in mediating the response of intestinal epithelial cells (IECs) to commensal-bacteria-derived signals[8]. Specifically, Hdac3 inactivation in the mouse intestinal epithelium decreased expression of genes associated with antimicrobial defence, altered the composition of the intestinal commensal bacteria and increased susceptibility to intestinal damage and inflammation. Notably, these effects were abrogated when these mice were re-derived under germ-free conditions[8]. Meanwhile, we have previously demonstrated a key role for HDAC3 in maintaining cell proliferation and survival, and inhibiting the differentiation of colon cancer cells[1,9–12].

HDAC3 has been reported to regulate lipid metabolism and homoeostasis in a number of tissues, although the directionality of the effect varies depending on the tissue. For example, Hdac3 inactivation in the liver increased the expression of genes that drive lipid synthesis and storage (e.g., Acab, Gpam, Elvol3 and Fasn) and causes hepatomegaly and fatty liver[13,14]. Typically, these genes are expressed in a circadian manner in hepatocytes, highly expressed at night when the animals are active and feeding, and repressed in a HDAC3-dependent manner during the day[13,14]. Deletion of Hdac3 resulted in failure of these genes to be repressed causing active lipid synthesis throughout the day. Similarly, conditional Hdac3 deletion in the heart resulted in mice developing severe hypertrophic cardiomyopathy, and the animals died within a few weeks when fed a high-fat diet[15]. In this tissue, Hdac3 inactivation resulted in downregulation of fatty acid oxidation genes, including Cpt1b, Ehhadh, Acads, Acad8, Crot and Acls5, causing the myocardium to become less efficient at oxidising lipid[15]. Finally, Hdac3 deletion in macrophages induced expression of Cd36, which improved the lipid efflux capacity of macrophages resulting in the formation of less vulnerable atherosclerotic lesions characterised by reduced lipid accumulation[16].

The intestinal epithelium plays a fundamental role in the uptake of lipid and its packaging into chylomicrons for delivery to peripheral tissues. In performing these functions, enterocytes consume up to 20% of all incoming energy, hence representing a major energy utilising tissue[17]. The intestinal epithelium also expresses high levels of a number of nuclear receptors, including members of the PPAR family[18], that play a key role in regulating the expression of lipid metabolism genes[19], and whose transactivation is regulated by HDAC3-containing co-repressor complexes in other tissues[20–22]. Despite these links, the role of HDAC3 in regulating lipid metabolism in the intestinal epithelium has not been fully investigated.

Herein, we unveil a previously unknown role for HDAC3 in regulating fatty acid oxidation in the small intestinal epithelium. We demonstrate that deletion of Hdac3 in intestinal epithelial cells (IECs) results in reduced adiposity in mice. The phenotype manifests progressively and is maintained over the entire lifespan of the mice. We demonstrate that deletion of Hdac3 in IECs results in coordinate induction of lipid oxidation genes and proteins, an increased rate of fatty acid oxidation, and extensive remodelling of the lipidome through de-repression of a PPAR-regulated transcriptional programme. Consistent with the effects induced by genetic inactivation of Hdac3, pharmacological inhibition of HDAC3 in C57BL/6 WT mice with the HDAC3-specific inhibitor RGFP966 increased expression of lipid oxidation genes in intestinal organoids and enterocytes in vivo. These findings establish HDAC3 as an orchestrator of lipid oxidation in enterocytes and a potential target for preventing obesity and its related complications.

## Results

**Intestinal deletion of Hdac3 reduces adiposity**. To investigate the role of HDAC3 in the intestinal epithelium, we generated intestinal-specific Hdac3 knockout (Hdac3[IKO]) mice by crossing Hdac3[lox/lox] with Villin[Cre] mice. Targeted-deletion of Hdac3 in the small intestinal epithelium was confirmed by genotyping, and at the mRNA and protein level (Supplementary Fig. 1a–c). Although requiring several weeks to manifest, a striking phenotype of Hdac3[IKO] mice was reduced weight gain, which was evident in both males and females. In mice maintained on a standard diet, the body weight difference was first evident at 11 weeks, and became increasingly prominent as mice were aged over 2 years (Fig. 1a, b). The nose-anus length was similar to Hdac3[WT] mice, suggesting that there was no growth defect (Fig. 1c). Rather, CT imaging revealed that the difference in body mass was due to a marked decrease in adiposity in Hdac3[IKO] mice (Fig. 1d, e), with no difference in lean body mass compared with Hdac3[WT] mice (Fig. 1f).

Consistent with the reduced adiposity, Hdac3[IKO] mice displayed improved glycemic control and insulin action as determined by oral glucose (Fig. 1g, j) and intra-peritoneal insulin tolerance tests (Fig. 1h, k), respectively. Hdac3[IKO] mice also had lower plasma triglycerides (TGs) following a 4 h fast, and TG levels remained consistently lower in Hdac3[IKO] mice compared with Hdac3[WT] mice following an oral lipid challenge (Fig. 1i, l). Comparatively, serum Tg levels were not different between Hdac3[WT] and Hdac3[IKO] mice fasted overnight, indicating these mice do not have an inherent difference in liver VLDL output (Supplementary Fig. 1d). Finally, the reduced adiposity also manifested in reduced liver steatosis and liver mass in Hdac3[IKO] mice at 12 months (Fig. 1m–o).

To determine if postnatal deletion of Hdac3 in the intestinal epithelium resulted in a similar phenotype, we generated Hdac3-Villin[Cre-ERT2] mice and induced Hdac3 deletion when mice reached 6 weeks of age by administering tamoxifen (Hdac3[IKO-IND]) (Supplementary Fig. 2a, b). While body weight was similar between groups prior to Hdac3 deletion, a reduction in body weight in Hdac3[IKO-IND] mice began to emerge ~11 weeks after Hdac3 deletion, which became progressively more pronounced over the 34-week experimental period (Supplementary Fig. 2c). The reduced body mass in Hdac3[IKO-IND] mice was due to a significant decrease in fat mass, but not lean mass (Supplementary Fig. 2d, e), collectively confirming the effects observed in Hdac3[IKO] mice. To control for possible CRE-mediated effects, the body weight of Villin[Cre] and Villin[Cre-ERT2] mice and WT littermates were monitored for 34 and 30 weeks, respectively, which revealed no difference compared with WT littermates

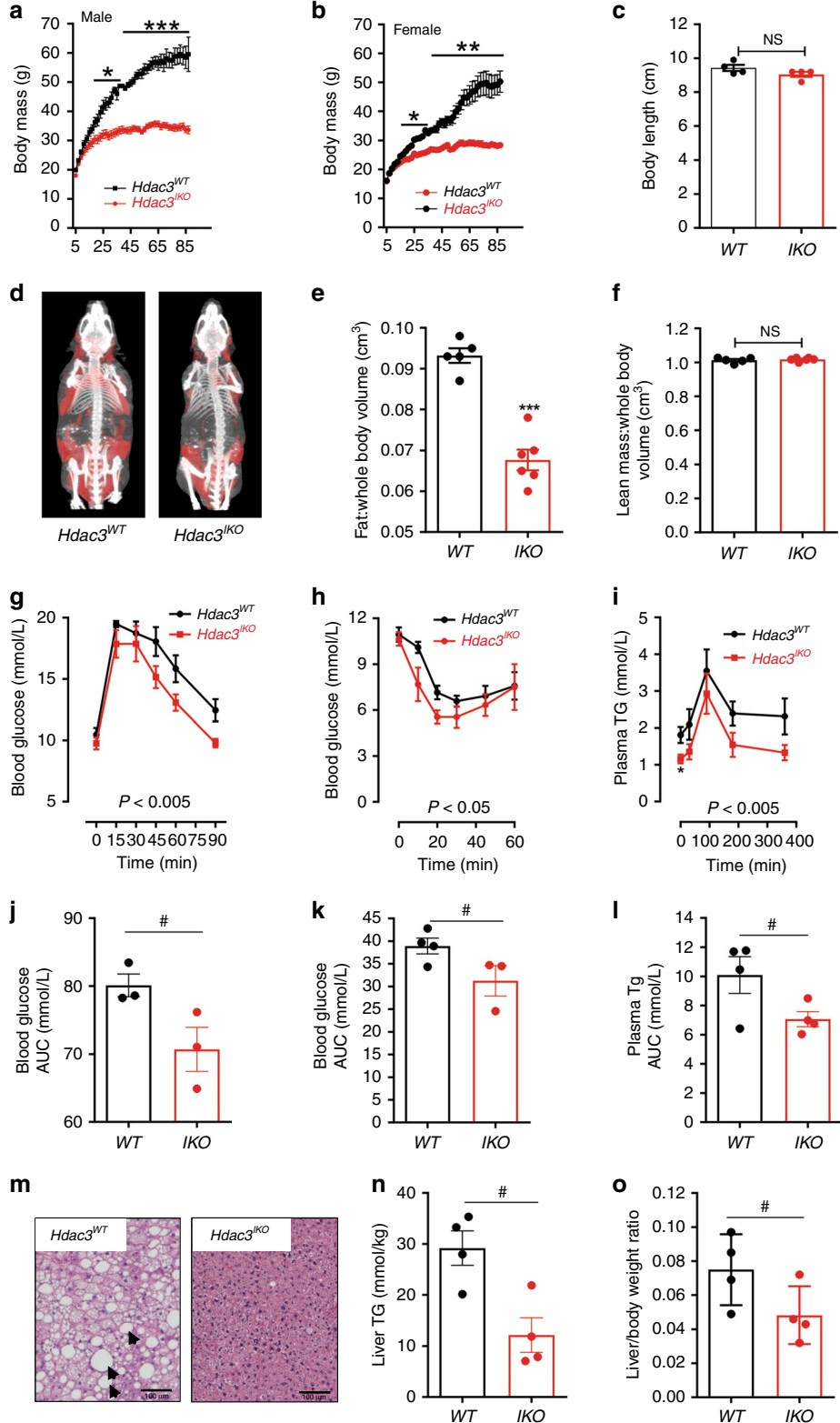

(Supplementary Fig. 3a–d). Furthermore, no difference in fasting plasma glucose, cholesterol, TG or insulin were observed between $Villin^{CreER}$ and $WT$ littermates treated with tamoxifen (Supplementary Fig. 3e–l), indicating there is no overt metabolic phenotype induced by CRE overexpression in the intestine, or tamoxifen treatment, and that the body weight differences observed in $Hdac3^{IKO}$ and $Hdac3^{IKO-IND}$ mice are due exclusively to loss of HDAC3.

Notably, the reduced weight gain of $Hdac3^{IKO}$ mice was not due to differences in physical activity (Fig. 2a), food intake (Fig. 2b) or faecal output (Fig. 2c). Furthermore, assessment of the energy density of stool collected from WT and $Hdac3^{IKO}$ mice by bomb calorimetry revealed no difference, demonstrating the body weight difference is not due to malabsorption (Fig. 2d). Comparatively, indirect calorimetry analysis revealed a significant increase in energy expenditure in $Hdac3^{IKO}$ mice (Fig. 2e).

**Fig. 1** Deletion of intestinal HDAC3 reduces adiposity. **a**, **b** Body weight of (**a**) male and (**b**) female $Hdac3^{WT}$ and $Hdac3^{IKO}$ mice determined from 5 to 85 weeks. **c** Body length of 6-month-old $Hdac3^{WT}$ and $Hdac3^{IKO}$ male mice. **d** Representative merged images of skeleton and fat (red) of 6-month-old male $Hdac3^{WT}$ and $Hdac3^{IKO}$ mice analysed by CT-scan and MRI. **e**, **f** Corresponding ratio of (**e**) fat/whole-body volume, and (**f**) lean mass/whole-body volume in $Hdac3^{IKO}$ and $Hdac3^{WT}$ mice. Values shown for **a–c** and **e–f** are the mean ± SEM of $n = 4$ and n=6 mice per genotype, respectively. All mice were fed a standard chow diet, and groups were compared using an unpaired $t$ test, $*P < 0.05$, $**P < 0.005$, $***P < 0.0005$. **g–l** Biochemical response of 6-month-old $Hdac3^{WT}$ and $Hdac3^{IKO}$ male mice fed a standard diet to (**g**, **j**) oral glucose challenge, (**h**, **k**) intra-peritoneal insulin challenge and (**i**, **l**) oral lipid (triglyceride) challenge. Values shown are mean ± SEM of $n = 4$ mice per group. **l** Plasma Tg levels were also compared at $t = 0$, using an unpaired $t$ test, $*P < 0.05$. For **g**, **h** and **i** groups were compared by two-way ANOVA, and for **j–l**, groups were compared using an unpaired $t$ test, $^{\#}P = 0.05$, one-tailed $t$ test, $*P < 0.05$, $**P < 0.005$, two-tailed $t$ test. **m** Liver histology of 12-month-old $Hdac3^{WT}$ and $Hdac3^{IKO}$ male mice fed a standard diet shows reduced evidence of liver steatosis (arrows) in $Hdac3^{IKO}$ mice. Shown is a representative image from $n = 4$ mice per genotype. **n** Corresponding liver triglyceride content and (**o**) liver/body weight ratio in 12-month-old $Hdac3^{WT}$ and $Hdac3^{IKO}$ male mice. Values shown are mean ± SEM of $n = 4$ mice per genotype. Groups were analysed using an unpaired one-sided $t$ test, $^{\#}P < 0.05$.

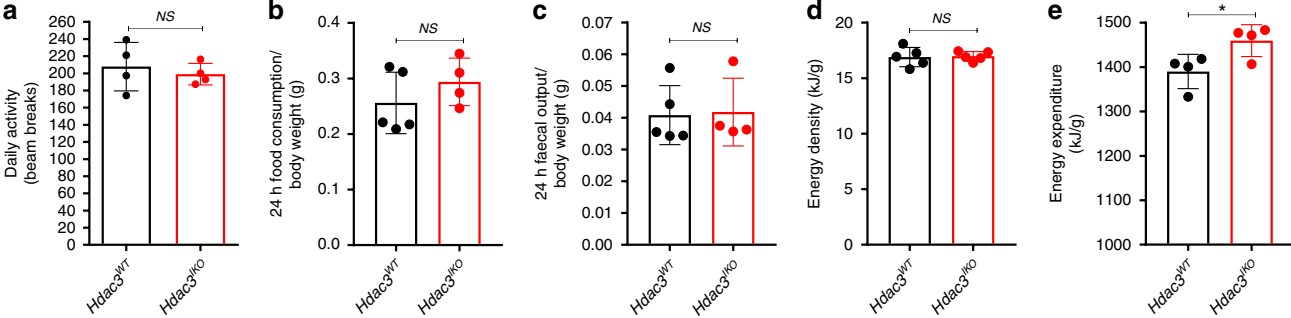

**Fig. 2** Reduced adiposity is not due to altered activity, food intake, faecal output or malabsorption. **a** Daily spontaneous physical activity, (**b**) food consumption, (**c**) faecal output and (**d**, **e**) energy content of stool in $Hdac3^{WT}$ and $Hdac3^{IKO}$ mice. **a** Spontaneous physical activity was monitored in 4–8-week-old male mice individually housed for 3 days and fed a standard chow diet. **b**, **c** Food consumption and faecal output measured in 4–8-week-old individually housed male mice over 5 consecutive days, and fed a standard chow diet. In all cases, values shown are the mean ± SEM of $n = 4$ (**a**, **b**, **e**) or n=5 (**c**, **d**) mice per genotype. $*P < 0.05$, NS not significant, unpaired $t$ test.

**Intestinal Hdac3 represses fatty acid oxidation genes**. To investigate potential mechanisms for the observed improvements in obesity, glycemic control and energy expenditure, we assessed the small intestinal epithelium for alterations in the lineage-specific differentiation of the major intestinal cell types. No significant differences in the abundance of enterocytes, goblet, Paneth or enteroendocrine cells were observed in the small intestine of $Hdac3^{WT}$ and $Hdac3^{IKO}$ mice (Supplementary Fig. 4a–h), suggesting the observed metabolic phenotype is not due to gross alterations in the maturation of these intestinal cell lineages.

We next performed gene expression profiling of intestinal epithelial cells (IECs) isolated from $Hdac3^{IKO}$ and $Hdac3^{WT}$ mice. Unsupervised cluster analysis of all genes clearly separated the samples according to genotype, demonstrating that $Hdac3$ deletion induces a marked transcriptional reprogramming of the intestinal epithelium (Fig. 3a). This included 878 genes that are upregulated and 1372 genes that are downregulated in $Hdac3^{IKO}$ mice (Fig. 3b, c; Supplementary Data 1).

KEGG pathway analysis of the upregulated transcripts identified enrichment of pathways regulated by members of the glutathione-S-transferase and cytochrome P450 gene families (metabolism of xenobiotics by $CYP450$, chemical carcinogenesis and glutathione metabolism), while the downregulated genes were enriched for regulators of calcium signalling and secretory and signalling processes dependent upon calcium signalling (Fig. 3d, e; Supplementary Table 1).

Genes upregulated in $Hdac3^{IKO}$ mice were also significantly enriched for multiple pathways related to lipid metabolism, including metabolic pathways, fatty acid degradation, biosynthesis of unsaturated fatty acids, fatty acid metabolism and PPAR signalling (Fig. 3d; Supplementary Table 1).

To validate these findings at the protein level, we performed proteomic profiling of IECs isolated from $Hdac3^{IKO}$ and $Hdac3^{WT}$ mice by LC-MS/MS. As for the gene expression analysis, unsupervised cluster analysis of the 1447 unique proteins detected in IECs separated the samples according to genotype, confirming marked reprogramming of the proteome in IECs from $Hdac3^{IKO}$ mice (Supplementary Fig. 5a, Supplementary Table 2). Of these, 94 proteins were significantly upregulated and 96 proteins significantly downregulated in $Hdac3^{IKO}$ mice. Furthermore, KEGG pathway analysis of the upregulated proteins revealed highly concordant overlap with the transcriptional profiling analysis, including significant enrichment of proteins involved in fatty acid metabolism, fatty acid degradation and PPAR signalling (Supplementary Fig. 5b, c).

More detailed investigation of the transcripts and proteins comprising these categories revealed coordinate upregulation of a number of regulators of mitochondrial ($Slc25A20$, $Acadl$, $Acads$, $Ech1$, $Hadhb$, $Acot1$, $Acot7$ and $Acot8$) and peroxisomal ($Ech1$, $Acaa1b$, $Hsd17b4$, $Ehhadh$, $Acox1$) β-oxidation (Fig. 3f, g).

**Hdac3 deletion remodels the lipidome of enterocytes**. To determine if the altered expression of genes and proteins involved in mitochondrial and peroxisomal fatty acid oxidation increases the rate of fatty acid oxidation in IECs in $Hdac3^{IKO}$ mice, we assessed fatty acid oxidation using a radiolabelled oleate tracer in segments of duodenum isolated from $Hdac3^{WT}$ and $Hdac3^{IKO}$ mice. While fatty acid uptake was similar between genotypes (Fig. 4a), the percentage of fatty acid oxidised was 2.3-fold higher in intestinal segments from $Hdac3^{IKO}$ mice. Repetition of these experiments in isolated enterocytes revealed an even higher 3.7-fold increase in the percentage of fatty acid oxidised in $Hdac3^{IKO}$ compared with $Hdac3^{WT}$ mice (Fig. 4b). This effect was specific

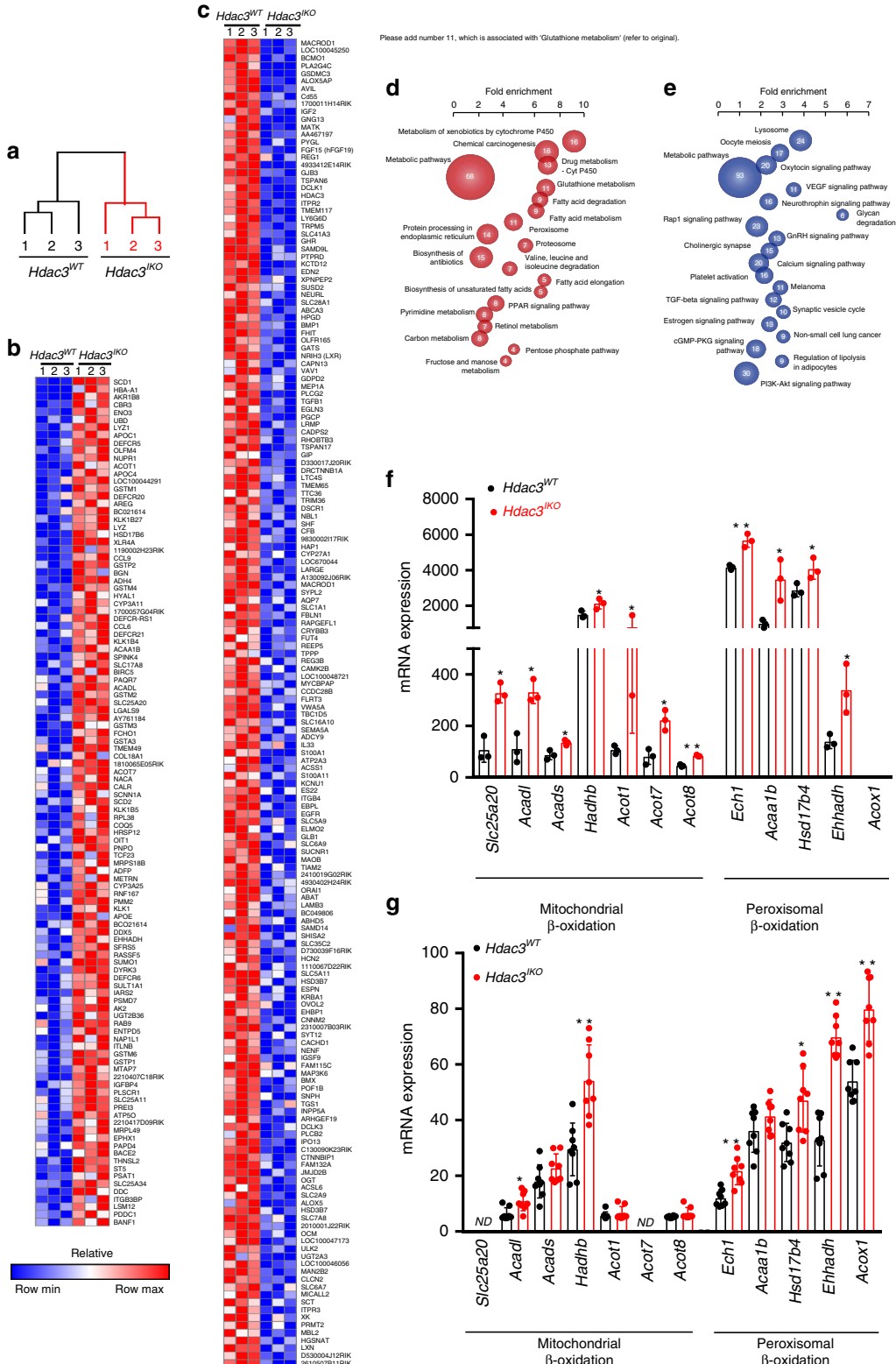

**Fig. 3** Deletion of intestinal HDAC3 induces fatty acid gene expression in enterocytes. Transcriptomic profiling of IECs isolated from 4–6-week-old *Hdac3WT* and *Hdac3IKO* mice fed a standard chow diet. **a** Unsupervised clustering of samples based on expression of all genes in IECs isolated from *Hdac3WT* and *Hdac3IKO* mice determined by illumina microarray analysis. **b** Genes upregulated and (**c**) genes downregulated in IECs isolated from *Hdac3WT* and *Hdac3IKO* mice. **d**, **e** Gene set enrichment analysis of transcripts differentially expressed between IECs isolated from *Hdac3*WT and *Hdac3IKO* mice. **f** mRNA and **g** corresponding protein expression of mitochondrial and peroxisomal β-oxidation genes in *Hdac3WT* and *Hdac3IKO* mice determined by transcriptomic (n = 3 per group) and proteomic profiling (n = 8 per group) using LC-MS/MS, respectively. ND not detected. In all cases, groups were compared using an unpaired t test, *P < 0.05, **P < 0.005.

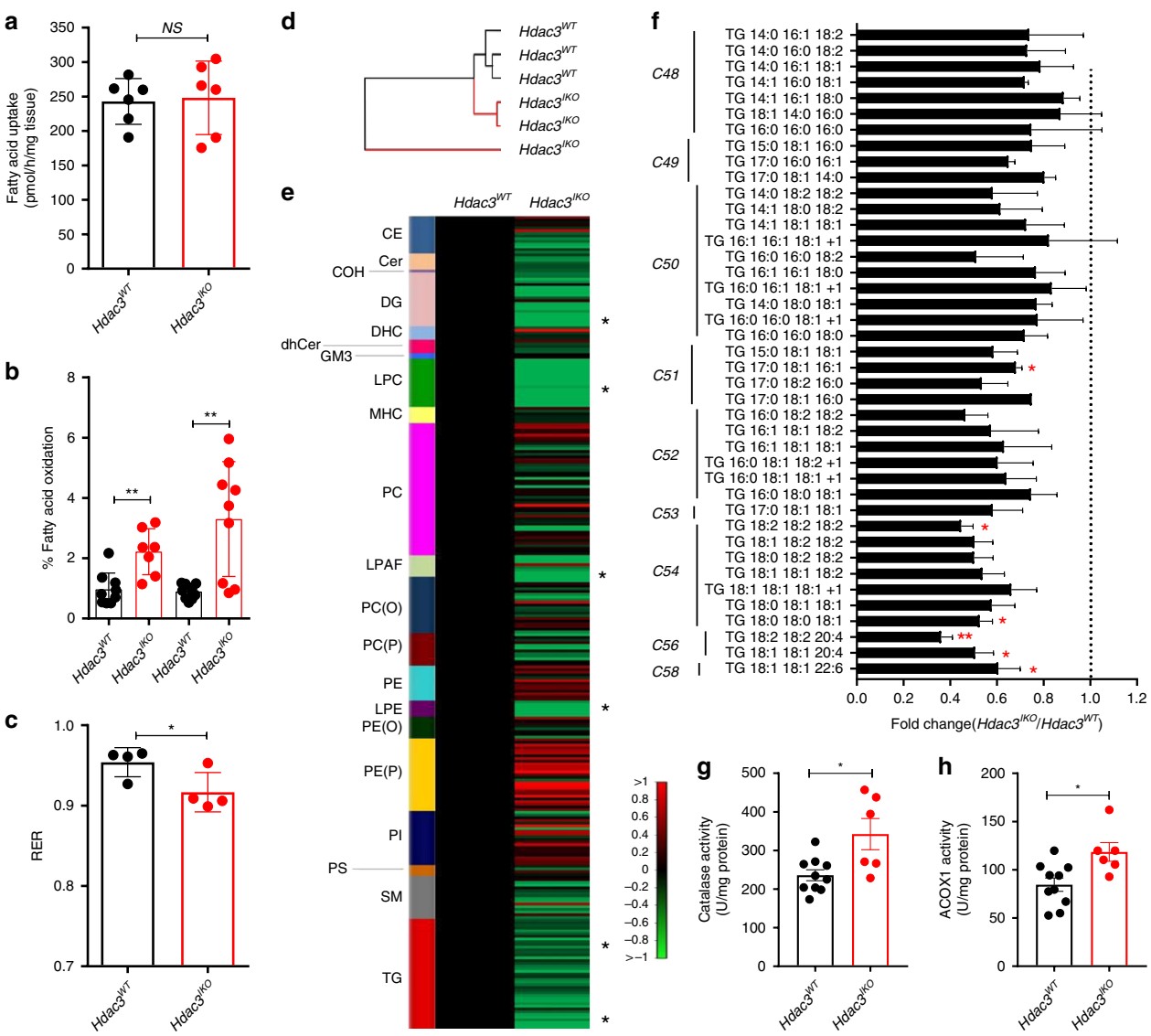

**Fig. 4** Deletion of HDAC3 increases lipid oxidation and remodels the lipidome of enterocytes. **a** Fatty acid uptake and (**b**) fatty acid oxidation assessed ex vivo in segments of the duodenum isolated from the small intestine of 4–6-week-old $Hdac3^{WT}$ and $Hdac3^{IKO}$ male mice fed a standard diet. Values shown are mean ± SD of individual tissue preparations generated from $n = 6$ mice (**a**) and $n = 3$ mice in triplicate (**b**) (NS not significant, $**P < 0.005$, unpaired $t$ test). **c** Respiratory exchange ratio (RER) measured over 24 h in 4–6-week-old $Hdac3^{WT}$ and $Hdac3^{IKO}$ male mice fed a standard diet. Mice were individually housed and the RER (VO2/VCO2) determined by indirect calorimetry. Values shown are mean + SD of $n = 4$ mice, $*P < 0.05$, unpaired $t$ test. **d**, **e** Profiling of the lipidome of IECs isolated from the entire small intestine of 4–6-week-old male $Hdac3^{WT}$ and $Hdac3^{IKO}$ mice fed a standard chow diet. **d** Unsupervised cluster analysis of samples based on abundance of 302 lipid species profiled in IECs isolated from $Hdac3^{WT}$ and $Hdac3^{IKO}$ mice. **e** Differential abundance of lipid species in IECs isolated from male $Hdac3^{WT}$ and $Hdac3^{IKO}$ mice. Data shown in the heatmap are the mean of $n = 3$ mice per genotype, $*P < 0.05$, unpaired $t$ test. **f** Fold change in individual triglyceride (TG) species of different aliphatic chain length in $Hdac3^{IKO}$ mice. Values shown are mean ± SEM of $n = 3$ mice per genotype, $*P < 0.05$, unpaired $t$ test. **g**, **h** Catalase (**g**) and ACOX (**h**) enzymatic activity in IECs isolated from the entire small intestine from 4–6-week-old mice fed a standard chow diet. Values shown are mean ± SD of individual tissue preparations generated from $n = 10$ $Hdac3^{WT}$ and $n = 6$ $Hdac3^{IKO}$ mice, $*P < 0.05$, unpaired $t$ test.

to the intestine as no genotype-related differences in fatty acid oxidation were observed in isolated skeletal muscle or liver (Supplementary Fig. 6a, b). To determine if the increased rate of fatty oxidation in the intestinal epithelium of $Hdac3^{IKO}$ mice was reflected in the whole animal, the respiratory exchange ratio (RER) was calculated by indirect calorimetry. The RER was significantly lower in $Hdac3^{IKO}$ mice, indicating $Hdac3^{IKO}$ mice utilise more fat as a fuel source compared to $Hdac3^{WT}$ mice (Fig. 4c).

To determine if the increased expression of fatty acid oxidation genes and increased rate of fatty acid oxidation in IECs manifest

in alterations to the lipidome, we profiled 302 independent lipid species in IECs isolated from $Hdac3^{WT}$ and $Hdac3^{IKO}$ mice using LC ESI-MS/MS. Consistent with the increased expression of fatty acid oxidation genes and rate of fatty acid oxidation, the lipidome of IECs in $Hdac3^{IKO}$ mice revealed extensive remodelling as demonstrated by unsupervised hierarchical clustering, which separated the samples according to genotype (Fig. 4d). In total, 40 lipid species were significantly differentially abundant between $Hdac3^{WT}$ and $Hdac3^{IKO}$ mice, with 39 present in reduced levels in $Hdac3^{IKO}$ mice. The majority of these lipid species comprised lysophosphotidyl cholines (LPCs), lysophosphotidyl

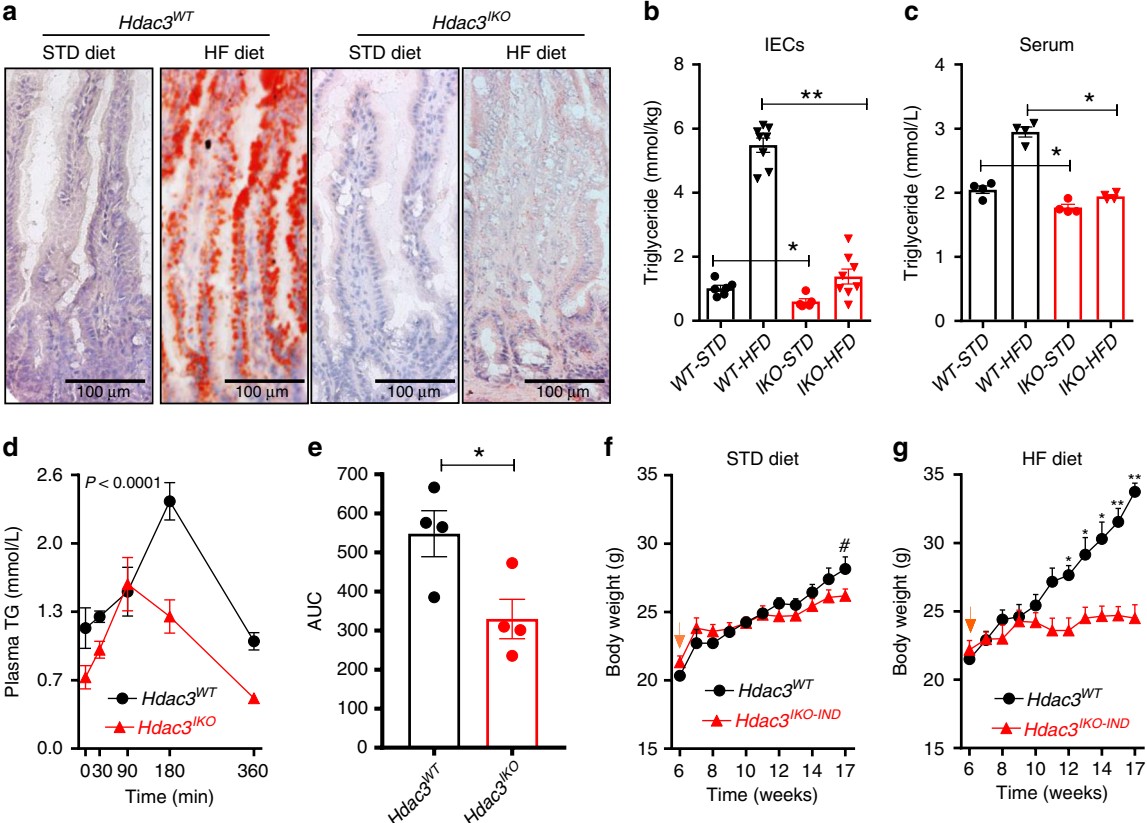

**Fig. 5** HDAC3 deletion reduces enterocytic triglycerides and protects against diet-induced obesity. **a** Oil Red staining of neutral lipids in the intestinal epithelium of 8–12-week-old *Hdac3*[WT] and *Hdac3*[IKO] male mice fed a standard (STD) or high-fat diet (HFD) for 4 weeks. **b**, **c** Triglyceride levels in (**b**) IECs or (**c**) serum from these mice. Values shown are mean ± SEM of $n = 6$ for mice fed a standard diet and $n = 8$ for mice fed a HFD, *$P < 0.05$, **$P < 0.005$, unpaired *t* test. **d** Lipid tolerance test of *Hdac3*[WT] and *Hdac3*[IKO] male mice fed a HFD for 4 weeks, and (**e**) corresponding area under the curve calculation. Values shown are mean ± SEM of $n = 4$ mice per genotype. **f**, **g** Difference in weight gain of 4–6-week-old *Hdac3*[WT] and *Hdac3*[IKO-IND] male mice fed a (**f**) standard (STD) or (**g**) HFD for 11 weeks. Mice were injected with tamoxifen at 6 weeks (orange arrow) to induce *Hdac3* deletion and body weight monitored for the following 11 weeks. Values shown are mean ± SEM of $n = 4$ mice each genotype. *$P < 0.05$, **$P < 0.005$, *t* test, #$P < 0.05$, one-sided unpaired *t* test.

ethanolamines (LPEs), diglycerides (DGs) and triglycerides (TGs) (Fig. 4e). Notably, peroxisomes are the primary site of β-oxidation of fatty acids with longer chain lengths (>22 carbons)[23]. Consistent with the increased expression of genes and proteins which regulate peroxisomal β-oxidation in *Hdac3*[IKO] mice, TGs with longer cumulative acyl chain lengths (C54, C56 and C58) were preferentially decreased in IECs from *Hdac3*[IKO] mice (Fig. 4f). To further investigate the increase in peroxisomal β-oxidation, we measured the enzyme activity of catalase and ACOX1 as readouts of peroxisomal activity, both of which were significantly higher in IECs isolated from *Hdac3*[IKO] mice (Fig. 4g, h).

Collectively, these findings demonstrate that deletion of *Hdac3* in IECs results in increased expression of genes and proteins involved in fatty acid oxidation, an increased rate of fatty acid oxidation in this tissue, and decreased abundance of particularly very long-chain triglycerides.

**Hdac3 deletion protects against diet-induced obesity.** We next tested the consequences of intestinal-specific *Hdac3* deletion on response to feeding a HFD. Feeding a HFD to *Hdac3*[WT] mice markedly increased neutral lipid accumulation in IECs as determined by Oil Red O staining (Fig. 5a) and biochemical assessment of TG levels (Fig. 5b). In comparison, lipid accumulation in

IECs was significantly attenuated in *Hdac3*[IKO] mice fed the same HFD (Fig. 5a, b). Furthermore, serum TG levels were significantly lower in *Hdac3*[IKO] mice fed a HFD (Fig. 5c), and serum TG was significantly reduced in these mice during an oral lipid tolerance test (Fig. 5d). Notably, the reduced levels of TGs in IECs in *Hdac3*[IKO] mice fed a HFD diet was not due to increased lipid excretion, as TG levels in the stools were not elevated compared to *Hdac3*[WT] mice (Supplementary Fig. 7a), and bomb calorimetry analysis of stool collected from *Hdac3*[WT] and *Hdac3*[IKO] mice revealed no difference in caloric content (Supplementary Fig. 7b). Finally, expression of genes required for intestinal fatty acid uptake and intracellular processing (*Cd36*, *Fabp*1-7, *Slc27A*1-6) or secretion (*Mttp*, *Apoa1*, *Apoa2 Apoa4*) were not different between genotypes (Supplementary Fig. 7c), suggesting the differences in lipid tolerance are unlikely to be mediated by differences in enteric lipid absorption or secretion.

Finally, we determined whether deletion of *Hdac3* in IECs protects against HFD-induced obesity. For this purpose, we utilised the *Hdac3*[IKO-IND] model, with *Hdac3* deletion induced at 6 weeks. No difference in body weight was observed prior to *Hdac3* deletion in mice fed either a standard (STD) or a HFD. Consistent with the findings in *Hdac3*[IKO] mice (Supplementary Fig. 2c), a modest but significant reduction in weight gain began to emerge in *Hdac3*[IKO-IND] mice fed a STD diet over the 11 week experimental period (Fig. 5e). This effect was markedly enhanced

in mice fed a HFD, with a significant difference in body weight evident 6 weeks after *Hdac3* deletion, establishing that intestinal *Hdac3* is required for weight gain, particularly when fed a high-fat diet (Fig. 5e, f).

**HDAC3-regulated lipid oxidation genes are PPAR targets**. The gene set enrichment analyses of transcripts (and proteins) increased in *Hdac3*[IKO] mice revealed significant enrichment of the "PPAR signalling pathway" (Fig. 3d; Supplementary Fig. 5b, Supplementary Table 2), which included a number of mitochondrial and peroxisomal fatty acid oxidation genes. All three members of the PPAR family of nuclear receptors (PPARα, δ and γ) are highly expressed in the intestinal epithelium[18], are established transcriptional regulators of lipid metabolism genes[24], and are known to recruit the HDAC3/NCoR1 co-repressor complex to mediate transcriptional repression in other tissues[21,25]. We therefore postulated that the coordinate induction of mitochondrial and peroxisomal β-oxidation genes in *Hdac3*[IKO] mice may be mediated through de-repression of a PPAR/HDAC3-regulated transcriptional programme. To address this, we first interrogated an independent microarray data set in which *WT* and *PPARα*[−/−] mice were treated with the PPARα agonist WY-14643, and gene expression changes profiled in the intestinal epithelium[26]. As shown in Supplementary Fig. 8a, many of the lipid oxidation genes upregulated upon *Hdac3* deletion, particularly genes involved in peroxisomal lipid oxidation, were selectively induced by WY-14643 in WT, but not *PPARα*[−/−] mice.

To address the involvement of other nuclear receptors, small intestinal epithelial organoids were generated from WT mice and stimulated with equimolar concentrations of agonists of PPARα (WY-14643), PPARβ/δ (GW501516) and PPARγ (Rosiglitazone), as well as agonists of FXR (GW4064) and LXR (T0901317). The PPARα agonist WY-14643 stimulated > twofold induction of 6/12 mitochondrial and peroxisomal β-oxidation genes, which were upregulated in *Hdac3*[IKO] mice (Supplementary Fig. 8b). The PPARδ agonist GW501516 also induced 3/12 of these genes, indicating the induction of lipid oxidation genes in *Hdac3*[IKO] mice is likely mediated by multiple PPARs (Supplementary Fig. 5c). Comparatively, the PPARγ agonist rosiglitazone, and agonists of FXR and LXR only minimally induced expression of these lipid oxidation genes (Supplementary Fig. 8d–f), although induction of some genes was evident.

Next, to determine whether HDAC3 is recruited to the promoters of these genes by PPARs, we first searched for putative PPAR response elements (PPREs) in these gene promoters by interrogating regions 1 kb up and downstream of the transcription start site. Putative PPREs were identified in *Acox1*, *Acaa1b*, *Acot1* and *Acot8* (Fig. 6a; Supplementary Table 4), of which the PPRE in the *Acox1* gene promoter has previously been shown to directly bind PPARα in hepatocytes[27]. To determine if HDAC3 is recruited to these sites, HDAC3 chromatin immunoprecipitation experiments were performed using freshly isolated IECs from wild-type mice, with IECs from *Hdac3*[IKO] mice serving as controls for non-specific antibody binding. HDAC3 binding to the PPRE in *Acox1* and the putative PPRE's in *Acaa1b*, *Acot1* and *Acot8* was significantly increased in *Hdac3*[WT] compared with *Hdac3*[IKO] mice, confirming recruitment of HDAC3 to PPREs in fatty acid oxidation genes in IECs (Fig. 6b). These findings establish a model whereby HDAC3 represses expression of fatty acid oxidation genes in a PPAR-dependent manner in the intestinal epithelium, and that HDAC3 inactivation results in de-repression and constitutive expression of these genes in vivo (Fig. 6c).

To further test this model, we predicted that PPAR agonists would preferentially induce expression of lipid oxidation genes in *Hdac3*[IKO] mice, due to absence of the HDAC3 co-repressor. Indeed, feeding *Hdac3*[WT] and *Hdac3*[IKO] mice a diet supplemented with 0.1% of the PPARα agonist WY-14643 for 24 h, resulted in preferential induction of lipid oxidation genes, particularly those involved in peroxisomal β-oxidation, in *Hdac3*[IKO] mice (Fig. 7a; Supplementary Fig. 9). Similar effects were observed at the protein level (Fig. 7b). Expression of SCD1, a known PPARα target gene[28], was also examined as an additional control and was preferentially induced by WY-14643 in *Hdac3*[IKO] mice (Fig. 7b).

To determine if *Hdac3*[IKO] mice were more susceptible to PPARα agonist-induced weight loss, *Hdac3*[WT] and *Hdac3*[IKO] mice were treated with 0.1% WY-14643 for 8 days and their body weight monitored. WY-14643 reduced weight gain in both *Hdac3*[WT] and *Hdac3*[IKO] mice over this experimental period, however, the reduction in weight gain was significantly more pronounced in *Hdac3*[IKO] mice, demonstrating their increased sensitivity to PPARα agonist treatment (Fig. 7c). Notably, food intake was increased in *Hdac3*[WT] mice fed WY-14643, and further increased in *Hdac3*[IKO] mice (Fig. 7d), indicating weight loss was not due to reduced food consumption.

**The HDAC3 inhibitor RGFP966 induces lipid oxidation genes**. Finally, to determine whether the transcriptional changes induced by *Hdac3* deletion could be phenocopied by pharmacological inhibition of HDAC3, organoids from C57BL/6 WT mice were treated with the HDAC3 inhibitor RGFP966, which revealed that 8 of the 12 mitochondrial and peroxisomal oxidation genes induced in *Hdac3*[IKO] mice were significantly induced in organoids treated with RGFP966 (Fig. 8a). To confirm these findings in vivo, WT mice were treated with RGFP966 in their diet for 8 days and induction of the β-oxidation gene programme examined in enterocytes. As observed in organoids, RGFP966 significantly induced expression of 9/12 of the mitochondrial and peroxisomal β-oxidation genes in enterocytes in vivo (Fig. 8b).

## Discussion

Here we describe a key role for the transcriptional co-repressor HDAC3 in the regulation of lipid oxidation in the intestinal epithelium. We demonstrate that intestinal-specific deletion of *Hdac3* results in the coordinate induction of multiple genes involved in mitochondrial and peroxisomal β-oxidation of fatty acids, which is associated with an enhanced capacity of IECs to oxidise fatty acids, and marked remodelling of their lipidome, particularly a reduction in the abundance of triglycerides and very long-chain fatty acids, under both standard and high-fat diet-fed conditions. In parallel, mice with intestinal HDAC3 deletion show a pronounced reduction in adiposity and are particularly resistant to high-fat diet-induced obesity, which is further associated with improved glycaemic control.

HDAC3 is a key component of the HDAC3-NCoR co-repressor complex, and our findings are consistent with the observation that deletion of other components of this complex in IECs also impacts oxidative capacity and expression of genes involved in lipid uptake and metabolism[29]. Specifically, intestinal-specific deletion of *Ncor1*, which performs a key scaffolding function of the HDAC3-NCoR complex and mediates recruitment by nuclear receptors via its RID domain, results in altered expression of a remarkably similar set of genes to those induced in *Hdac3*[IKO] mice, including genes involved in fatty acid metabolism and PPAR signaling[29]. Our finding of a central role for HDAC3 in regulating expression of fatty acid metabolism genes in the intestine is also consistent with its role in lipid metabolism in other tissues including the liver, heart, muscle and inflammatory cells. However, the specific genes regulated, and the

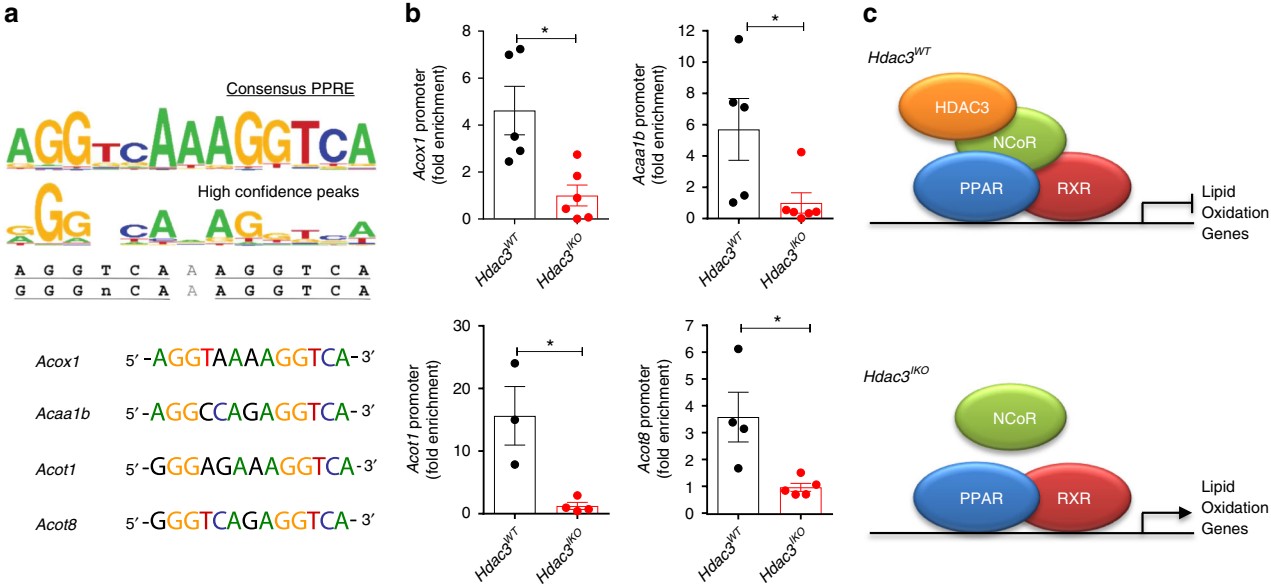

**Fig. 6** HDAC3 regulates expression of PPAR-target genes. **a** Identification of putative PPAR response elements (PPREs) in the promoters of *Acox1*, *Acaa1b*, *Acot1* and *Acot8*, and comparison to the consensus PPRE. Consensus PPRE obtained from the JASPAR database[56]. **b** Chromatin immunoprecipitation (ChIP) analysis of HDAC3 occupancy at these putative PPREs in IECs isolated from 4–6-week-old *Hdac3*^WT and *Hdac3*^IKO male mice fed a standard chow diet. Values shown are mean + SEM of ChIP experiments performed on enterocytes from $n = 6$ biological replicates with outliers removed (see source data file), *$P < 0.05$, **$P < 0.005$, unpaired $t$ test. **c** Model of de-repression of fatty acid oxidation genes in *Hdac3*^IKO mice.

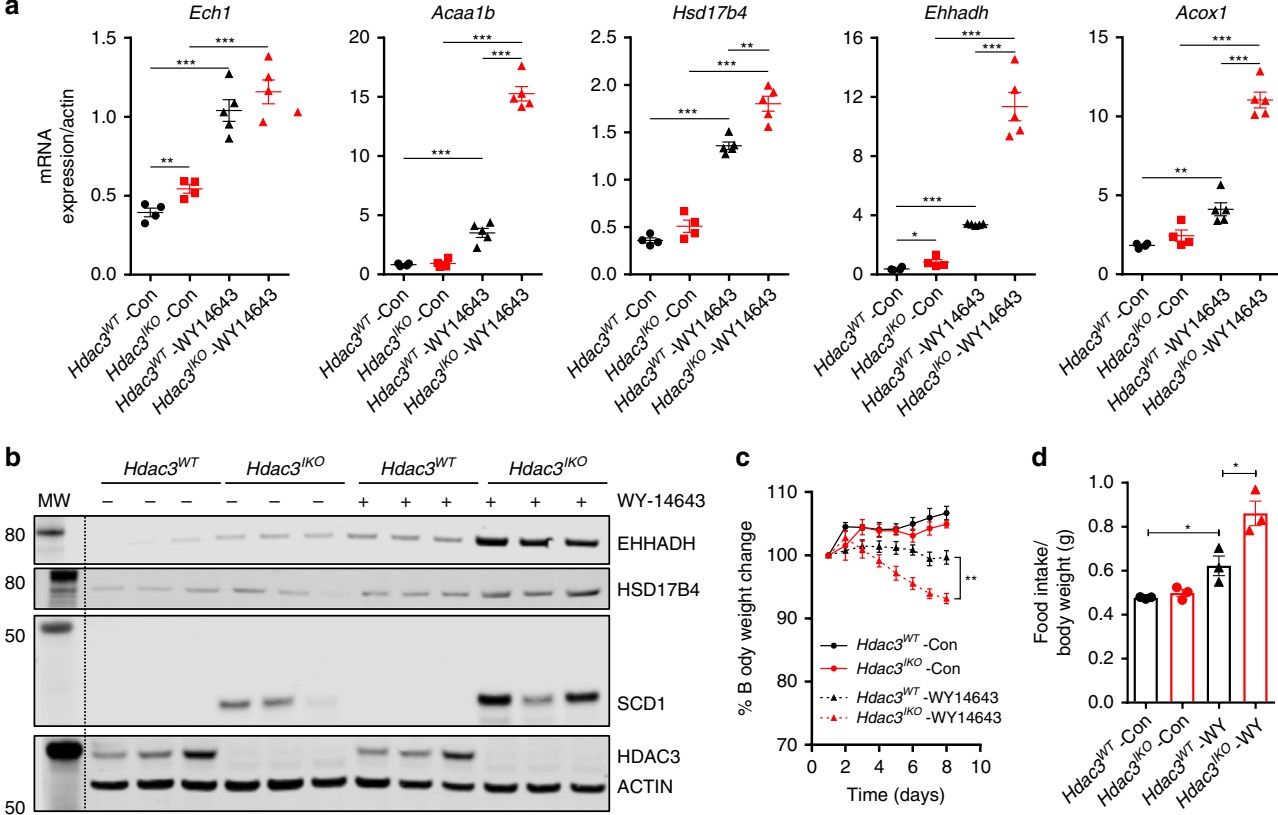

**Fig. 7** HDAC3 deletion enhances sensitivity to PPAR agonists. The PPARα agonist (WY-14643) preferentially induces fatty acid oxidation genes and reduces weight gain in *Hdac3*^IKO mice. **a**, **b** *Hdac3*^WT and *Hdac3*^IKO male mice 4–6 weeks of age were fed a standard chow diet supplemented with or without WY-14643 (WY) (0.1% w/w) for 24 h. Gene and protein expression changes in enterocytes isolated from the duodenum were determined by (**a**) qPCR and (**b**) western blot, respectively. Values shown are mean + SEM of $n = 4$ mice per treatment group, *$P < 0.05$, **$P < 0.005$, unpaired $t$ test. Western blots in (**b**) were performed on lysates isolated from three separate mice. **c**, **d** *Hdac3*^WT and *Hdac3*^IKO male mice (4–6 weeks of age) were fed a standard chow diet supplemented with or without WY-14643 (0.1% w/w) for 8 days, and (**c**) body weight and (**d**) average daily food intake measured over the final 3 days of the experiment. Values shown are mean + SEM of $n = 4$ mice per genotype, *$P < 0.05$, **$P < 0.005$, unpaired $t$ test.

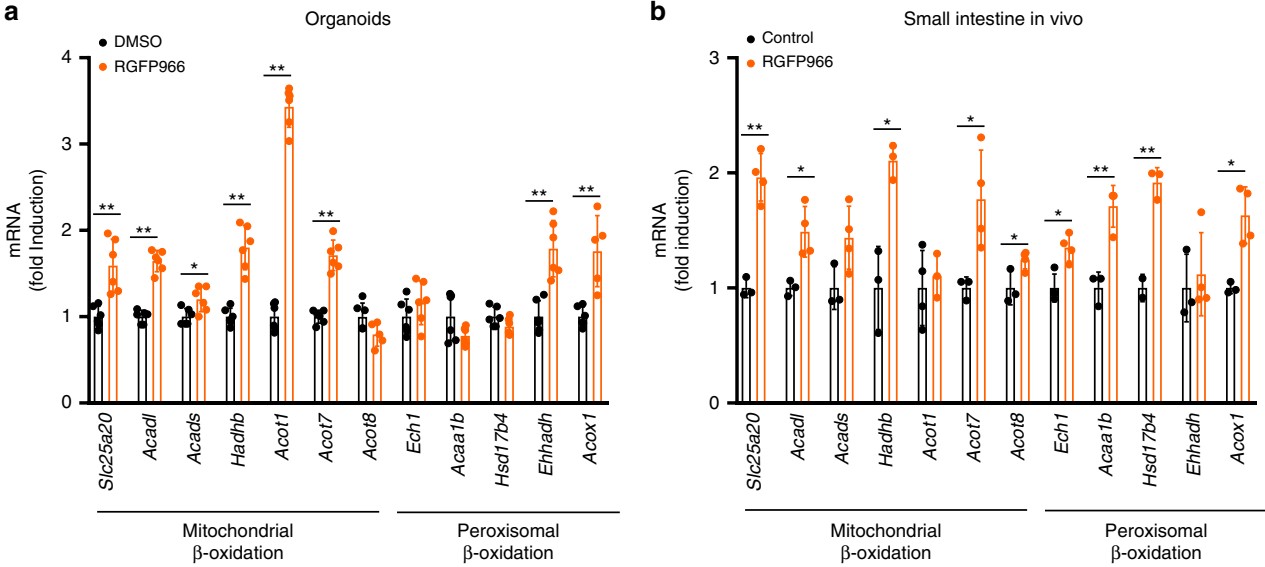

**Fig. 8** Pharmacological inhibition of HDAC3 induces lipid oxidation genes in enterocytes. The HDAC3 inhibitor (RGFP966) induces expression of fatty acid oxidation genes in small intestinal organoids in vitro and in enterocytes in vivo and reduces weight gain in WT mice. **a** Small intestinal organoids generated from 4–6-week-old C57BL/6 male mice fed a standard diet were treated with RGFP966 (100 μM) for 6 h, and gene expression changes determined by qPCR. Values shown are mean + SD or two independent experiments performed in triplicate. **b** C57BL/6 male mice 4–6 weeks of age were fed a standard chow diet supplemented with or without RGFP966 (150 mg/kg) for 8 days, and gene expression changes in the duodenum determined by qRT-PCR. Values shown in **a** and **b** are mean + SEM of $n = 4$ mice per treatment group, $*P < 0.05$, $**P < 0.005$, unpaired $t$ test.

consequent impact on fatty acid metabolism is highly tissue specific. For example, while our finding that lipid oxidation genes are induced upon *Hdac3* deletion in the intestine is similar to the effect observed in macrophages[16], *Hdac3* deletion in the liver[13], and heart[15], induces expression of genes required for fatty acid synthesis.

These differences are likely due to the differential recruitment of the HDAC3/NCoR co-repressor complex by transcription factors with tissue-specific expression patterns and functions. For example, the role of HDAC3 in repressing lipogenesis genes in hepatocytes is mediated through its recruitment by the *REV-ERB-α* nuclear receptor[30]. In comparison, we provide several lines of evidence to support that HDAC3 acts to repress genes involved in fatty acid oxidation in enterocytes by repressing transcriptional targets of the PPAR family of nuclear receptors. First, enrichment analyses of the genes and proteins induced in *Hdac3*[IKO] mice revealed significant enrichment of PPAR-target genes. Second, interrogation of an independent microarray data set of IECs from WT and *PPARα*[−/−] mice treated with WY-14643 revealed selective induction of the same mitochondrial and peroxisomal fatty acid β-oxidation genes in WT mice[26]. Third, several of the fatty acid oxidation genes upregulated in *Hdac3*[IKO] mice were similarly induced in intestinal organoids treated with PPAR agonists in vitro, and ChIP analyses demonstrated binding of HDAC3 to PPREs in these gene promoters in *Hdac3*[WT], but not *Hdac3*[IKO] mice. Finally, PPAR agonists induced more robust induction of the lipid oxidation genes in *Hdac3*[IKO] mice.

Importantly, these findings are consistent with several prior studies which have demonstrated that administering mice PPAR agonists such as fenofibrate[31], bezafibrate[32], WY-14643[33], diacylglycerol[34] or DHA[35] induces expression of lipid oxidation genes, and β-oxidation, in enterocytes[34,35], and concomitantly suppresses enterocytic lipid accumulation[33] and post-prandial lipidemia[32,35]. Furthermore, our findings are also consistent with the phenotype of mice with intestinal-specific deletion of PPARδ which are more prone to high-fat diet-induced obesity[36]. We now extend these findings to demonstrate that loss of the HDAC3 co-

repressor potentiates the effects of PPAR agonists on inducing expression of lipid oxidation genes in enterocytes, and weight loss.

*Hdac3* deletion in the intestine also resulted in reduced adiposity that manifested progressively over time. This was not due to reduced food intake or malabsorption, and was associated with reduced serum triglycerides, increased whole-body energy expenditure and increased systemic fat oxidation, raising the possibility that the increased oxidation of lipids in the intestine may contribute to the reduced adiposity phenotype.

However, the large number of cellular processes regulated by HDAC3 precludes the ability to directly link the increase in enterocytic lipid oxidation with reduced adiposity. For example, *Hdac3* deletion also induced expression of genes involved in xenobiotic metabolism as well as other aspects of lipid metabolism (e.g., *Scd1, Scd2*), which may be the result of deregulation of PPARs as well as other HDAC3-interacting nuclear receptors. HDAC3 also plays a key a role in integrating signals derived from the gut microbiome to alter enterocyte physiology[8,37], which may also impact on adiposity[38]. It is therefore likely that the induction of lipid oxidation in enterocytes, combined with these other effects, ultimately determines the body weight phenotype of *Hdac3*[IKO] mice. We also acknowledge that while expression of genes involved in lipid absorption and lipoprotein secretion were not altered in *Hdac3*[IKO] mice, additional studies are required to determine if *Hdac3* deletion in enterocytes may contribute to the metabolic phenotype by impacting on lipid or lipoprotein secretion.

Intestinal-specific *Hdac3* deletion was also previously shown to alter expression of both a common as well as a unique subset of genes in conventionally versus germ-free-housed mice, and to specifically predispose conventionally housed mice to intestinal inflammation and colitis[8]. Notably, this inflammatory phenotype was not observed in our cohort, which likely reflects differences in the microbiota between the two colonies. During the review of this paper, a follow up study from this group also reported reduced weight gain in *Hdac3*[IKO] mice, which was linked to

altered expression of *Chka, Mttp, Pck1, Apoa1*. Furthermore, an abnormal increase in lipid levels was observed in enterocytes in these mice[39], which the authors speculate may be driven by the induction of PCK1, which is predicted to increase glyceroneogenesis[39]. Consistent with likely differences in the microbiota, expression of *Chka, Mttp, Pck1* and *Apoa1* were not induced in our *Hdac3*[IKO] colony, which likely explains the different phenotypes observed. Comparatively, interrogation of the gene expression changes commonly altered by *Hdac3* deletion in conventionally housed and germ-free *Hdac3*[IKO] mice revealed enrichment of PPAR-target genes in both settings[8], demonstrating this is a direct HDAC3-regulated transcriptional programme.

An important finding of this study is the demonstration that the intestinal epithelium represents a potential point of intervention in lipid-related diseases. In this regard, inhibiting the capacity of the intestinal epithelium to absorb lipid using pancreatic lipase inhibitors, such as orlistat, has been previously explored and was effective at reducing obesity[40]. Intestinalspecific overexpression of the mitochondrial protein SIRT3 also increased metabolic activity and expression of ketogenic genes in enterocytes and protected mice from developing insulin resistance when fed a high-fat diet[41].

However, an important consideration in linking changes in enterocyte fatty acid oxidation with adiposity is the need to enhance our understanding of the contribution of enterocytes to whole-body fatty acid oxidation and energy expenditure. While some studies have estimated that up to 20% of total energy expenditure in humans occurs in the GI tract[17,42,43], a recent study demonstrated that increasing fatty acid oxidation specifically in enterocytes of mice by overexpressing a mutant form of carnitine palmitoyltransferase 1a, was not sufficient to alter body weight[44]. We note however that while CPT1 primarily regulates mitochondrial β-oxidation, our findings identify a role for HDAC3 in regulating both mitochondrial as well as peroxisomal fatty acid oxidation which may result in a more profound effect.

While we cannot with certainty define the mode of action for HDAC3 in the regulation of systemic adiposity, our findings clearly establish that inhibiting intestinal HDAC3 has the capacity for therapeutic utility in the treatment of obesity and metabolic co-morbidities. However, an important consideration in targeting HDAC3 is the contrasting effects of HDAC3 inactivation in different tissues, and potential toxicities that may be invoked as a result of systemic HDAC3 inhibition, particularly the liver, where HDAC3 inhibition promotes lipid synthesis and liver steatosis[13]. Strategies aimed at selective inhibition of HDAC3 in the intestinal epithelium, through targeted delivery methods or carefully optimised dosing strategies are therefore likely to be needed to explore this therapeutic opportunity. A further limitation of this approach is the possibility that increasing peroxisomal βoxidation may generate free radicals in enterocytes[45], which may have potentially deleterious consequences if not appropriately scavenged. Notably, expression and activity of the peroxisomal ROS scavenging enzyme catalase was significantly increased in *Hdac3*[IKO] mice. A consideration during long-term targeting of the HDAC3/PPAR axis therefore could be the parallel administration of ROS-scavengers to offset these effects. The relative efficacy of activating intestinal PPAR's versus inhibiting intestinal HDAC3 as a means of enhancing enterocyte oxidation, or even combination strategies targeting both proteins, would also be worthy of further investigation.

In summary, we demonstrate a key role for the transcriptional co-repressor *Hdac3* in the regulation of PPAR-regulated expression of lipid oxidation genes in the intestinal epithelium. We also demonstrate that intestinal-specific deletion of *Hdac3* protects against adiposity, suggesting HDAC3 may represent a potential therapeutic target in preventing obesity and related diseases.

## Methods

**Generation of Hdac3[IKO] and Hdac3[IKO-IND] mice.** *Hdac3*[Lox/Lox] mice which were generated previously[13,46,47] were maintained on the C57BL/6 background. *Villin*[Cre] mice on the C57BL/6 background which express *Cre* recombinase under the control of the intestinal-specific *Villin* promoter were obtained from Jackson Laboratories, and *Villin*[Cre-ERT2] mice were obtained from Dr. Sylvie Robine. *Hdac3*[Lox/Lox] mice were crossed to *Villin*[Cre] or *Villin*[Cre-ERT2] mice to obtain *Hdac3*[Lox/ Lox];*Villin*[Cre] (*Hdac3*[IKO]), *Hdac3*[Lox/Lox];*Villin*[Cre-ERT2] (*Hdac3*[IKO-IND]) and *Hdac3*[Lox/Lox]; *Villin*[WT] (*Hdac3*[WT]) mice. For *Hdac3*[Lox/Lox];*Villin*[Cre-ERT2] mice, deletion of *Hdac3* was induced by four consecutive intra-peritoneal injections with tamoxifen (10 mg/ml) when mice were 4–6-weeks old. *Hdac3*[Lox/Lox];*Villin*[Cre-ERT2] littermates injected with vehicle alone (sunflower oil), served as controls. We note that as the majority of experiments were performed in 4–8-week-old mice interpretation of findings are primarily applicable to this age group. To control for possible off target effects of *Cre*, 4–6-week-old *villin-Cre*[ERT] mice and WT littermates were injected with four consecutive intra-peritoneal injections of with tamoxifen (10 mg/ml).

Primers used for determining *Hdac3* genotype were F: CCACTGGCTTCTCCT AAGTTC; and R: CCCAGGTTAGCTTTGAACTCT. Primers used for genotyping *Villin*[Cre] transgenic mice were F: GTGTGGGACAGAGAACAAACC and R: ACA TCTTCAGGTTCTGCGGG. Primers used for genotyping *Villin*[Cre-ER] transgenic mice were F: CAAGCCTGGCTCGACGGCC and R: CGCGAACATCTTCAGGTTCT.

All experiments involving mice complied with the ethical regulations for animal testing and research outlined by the National health and medical research council (NHMRC) of Australia, and all procedures were approved by the Austin Health Animal Ethics Committee.

**Isolation of intestinal epithelial cells (IECs).** IECs were isolated from the entire small intestine (duodenum, jejunum and ileum) from *Hdac3*[IKO] and *Hdac3*[WT] mice by incubation of the small intestine in 15 mM EDTA for 15 min and pellets frozen. For samples to be used for gene expression analysis, the buffer was supplemented with RNA secure (Thermo Fisher Scientific, Waltham, MA, USA).

**Quantitative real-time PCR.** The total mRNA was isolated using the High Pure RNA isolation kit from Roche Diagnostics (Roche, Indianapolis, IA), and reverse transcription performed using the Superscript III cDNA synthesis kit from Invitrogen (Carlsbad, CA) using 500 ng of the total RNA. Quantitative real-time PCR was performed using the Fast Start Universal SYBR Green master mix from Roche, on a 7500 Fast Real Time PCR System (Applied Biosystems, Carlsbad, CA). Primers used are listed in Supplementary Table 3.

**Western blot analysis.** Protein isolation was performed with RIPA buffer (50 mM Tris-HCl pH 7.5, 150 mM NaCl, 1% NP-40, 0.5% sodium deoxycolate, 1 mM EDTA pH 8 and 1 tablet of protease inhibitor cocktail Roche Complete (Roche) and 1 tablet of phosphatase inhibitor PhosSTOP (Roche) added per 10 ml of buffer. In total, 40 μg of protein was resolved under denaturing conditions (SDS-PAGE) and transferred to the PVDF membrane, and the membrane was blocked in Odyssey blocking buffer (Li-Cor Bioscience). Blots were probed with anti-HDAC3 (cat no 2632, Cell Signalling Denver, CO, 1:1000 or Abcam, Cambridge UK, AB93172, 1:1000), anti-HSD17B4 (Novus Biologicals, Colorado, US, NPB1-33192, 1:2500), anti-SCD1 (Cell Signaling, CST2784s, 1:2500) at 4 °C overnight or anti-βactin (cat no A5316, Sigma-Aldrich, St. Louis, MO, 1: 10,000) at room temperature 1 h. Secondary antibodies used were IRDye® 800CW Goat anti-Mouse IgG (P/N: 926-32210, 1:10,000) and IRDye® 680RD goat anti-rabbit IgG (P/N: 926-68071, 1:10,000, Li-Cor Bioscience, Nebraska, USA). The infrared fluorescence image was obtained using Odyssey infrared imaging system (Li-Cor Bioscience). Uncropped blots are available in the source data file.

**Body composition analysis.** *Hdac3*[WT] and *Hdac3*[IKO] mice were imaged by single photon emission computed tomography (SPECT)/computed tomography (CT) and positron emission tomography (PET)/magnetic resonance imaging (MRI), using dedicated small animal nanoPET/MR and nanoSPECT/CT cameras (nanoScan®, Mediso, Budapest, Hungary). Images were acquired for 30 min, whilst animals were under anaesthetic. Image analysis was performed using the PMOD (PMOD Technologies LTD, Zurich, Switzerland) analysis software, by segmenting the CT and MRI images according to tissue density—first for total volume and then for fat volume[48].

Body composition of *Hdac3*[IKO-IND] mice was assessed using a dual-energy Xray absorptiometry (DEXA) scanner (Lunar PIXImus; GE Healthcare, Sydney, New South Wales, Australia). Scans were analysed using the PIXImus software (version 2.10) with the head excluded from the analysis.

**Glucose, insulin and lipid tolerance tests.** Oral glucose tolerance tests (OGTT) were performed by fasting mice for 4 h, followed by administration of glucose (2 g/ kg body weight) via oral gavage. Blood samples were collected from the tail vein, and glucose levels measured using an Accu-Chek Performa Blood Glucose Meter Device (Roche) at time 0 (pre-gavage) and 15, 30, 45, 60 and 90 min post glucose administration. For insulin tolerance tests, mice were fasted for 4 h and injected IP

with insulin (0.75 U/kg body weight). Blood samples were taken 10 min prior to insulin administration, at time 0, and 10, 20, 30, 45 and 60 min after insulin administration for measurement of glucose levels. For lipid tolerance tests, mice were fasted for 4 h and olive oil (10 µL/g of body weight) administered via oral gavage. Blood samples were collected by tail vein bleeds at time 0 (pre-gavage) and 30, 90, 180 and 360 min post olive oil administration and serum triglyceride levels measured using the Infinity Triglycerides Liquid Stable Reagent (Thermo Scientific).

**Faecal output and bomb calorimetry.** Faecal output was assessed in singly housed mice over 24 h by collection of all stool in the cage. The bedding was then replaced, and the collection repeated for 2 more days. Energy content in faeces was determined on combustion within a semi-micro oxygen combustion vessel (PARR 1109 A) and measurement by a bomb calorimeter (PARR 6200; John Morris Scientific Pty Limited, Australia). Energy equivalence of the vessel and bomb were determined with a benzoic acid standard.

**Indirect calorimetry studies.** To measure food intake, locomotor activity, oxygen consumption (VO$_2$), carbon dioxide elimination (VCO$_2$) and computation of the respiratory exchange ratio (RER), singly housed mice were analysed using CLAMS (Comprehensive Lab Animal Monitoring System, Columbus Instruments) for 3 days with food and water provided ad libitum. Animals were adapted for 2 days prior to commencing measurements.

**HDL and LDL profiling.** HDL and LDL were measured in plasma collected from mice fasted overnight and following refeeding for 2 h using colorimetric assays (HDL Cat#79990, LDL Cat#79980) from Crystal Chem (Grove Village, IL, USA), as per the manufacturer's instructions.

**Measurement of insulin levels in plasma.** Insulin levels were measured in plasma collected from mice fasted overnight and following refeeding for 2 h using an ELISA kit from Crystam Chem (Cat#90080) as per the manufacturer's instructions.

**Microarray analysis.** The total RNA was extracted from IECs as above, and 3 µg of RNA submitted to the Australian Genome Research Facility for probe preparation and microarray analysis using the MouseRef-8 v2.0 Gene Expression BeadChip from Illumina (San Diego, CA, USA).

**Immunohistochemistry.** Formalin-fixed paraffin embedded sections of mouse small intestine were deparaffinised and rehydrated through serial xylene and ethanol washes, then rinsed in tap water. Endogenous peroxidase was quenched by incubating the sections in 3% in H$_2$O$_2$ in RO water (Science Supply Associates, Australia) at room temperature for 10 min. Antigen retrieval was performed by incubation of the tissue in citric acid buffer at 100 °C for 30 min. Sections were stained with anti-chromogranin (Abcam, Cambridge, UK, ab85554, 1:500) or anti-Lyzozyme (Thermo Fisher Scientific, Rockford, IL, 1: 300). Anti-mouse and anti-rabbit secondary antibodies and detection system were from Dako (Glostrup, Denmark). In all cases sections from control and experimental groups were analysed in parallel. The stained slides were counterstained with haematoxylin, dehydrated through serial ethanol and xylene washes and mounted using DPX (Sigma-Aldrich, USA).

**Alkaline phosphatase and alcian blue staining.** Alkaline phosphatase (ALP) activity was determined using the Vector red alkaline phosphatase substrate kit I from Vector Laboratories (Burlingame CA), according to the manufacturer's instructions. Alcian blue staining was performed by incubation with 1% alcian blue solution in 3% acetic acid for 30 min and counterstaining in 0.1% nuclear fast red for 5 min[49].

**Proteomic profiling.** Total protein (250 µg) isolated from $n = 8$ mice per group was subjected to acetone precipitation, reduced by shaking for 30 min at 37 °C in 10 mM dithiothreitol (DTT) and alkylated using 55 mM iodoacetamide (IAA) in the dark for 45 min. Protein samples were subsequently digested with trypsin at 37 °C overnight using a protein to trypsin ratio of 50:1 (w/w), and purified by solid-phase extraction.

LC-MS/MS was carried out on a LTQ Orbitrap Elite (Thermo Scientific) with a nanoelectrospray interface coupled to an Ultimate 300 RSLC nanosytem (Dionex). The nanoLC system was equipped with an Acclaim Prepmap Nanotrap column (Dionex C18, 100 Å, 75 µm × 2 cm) and an Acclaim Pepmap analytical column (Dionex C18, 2 µm, 100 Å, 75 µm × 15 cm). The tryptic peptides were injected onto the enrichment column at an isocratic flow of 5 µL/min for 5 min in 2% v/v CH3CN containing 0.1% v/v formic acid, before the enrichment column was switched in-line with the analytical column. The eluents were 0.1% v/v formic acid (solvent A) and 100% v/v CH3CN in 0.1% v/v formic acid (solvent B). The flow gradient was (i) 0–5 min in 3% Solvent B, (ii) 5–6 min in 3–12% Solvent B, (iii) 6–26 min in 12–35% Solvent B, (iv) 26–28 min in 35–80% Solvent B, (v) 28–30 min in 80–80% Solvent B and (vi) 30–31 min in 80–83% Solvent B, followed by

equilibration in 3% Solvent B for 8 min before the next sample injection. The LTQ Orbitrap Elite mass spectrometer was operated in the data-dependent mode with a nano ESI-spray voltage of +1.9 kV, a capillary temperature of 250 °C and a S-lens RF value of 60%. A lockmass of 445.120025 was used. Spectra were acquired first in positive mode with full scan, scanning from m/z 300 to 1650 in the FT mode at 240,000 resolution followed by collision-induced dissociation in the linear ion trap. The twenty most intense peptide ions with charge states ≥2 were isolated and fragmented using a normalised collision energy of 35 and an activation Q of 0.25. Dynamic exclusion of 30 s was applied. Extract-MSn within Bioworks 3.3.1 (Thermo Scientific) was used to generate peak list files with the following parameters: minimum mass 300; maximum mass 5000; grouping tolerance 0 Da; intermediate scans 200; minimum group count 1; 10 peaks minimum and total ion current of 100. Peak lists for each LC-MS/MS run were merged into a single mascot generic file format for MASCOT searches. LC-MS/MS spectra were searched against the NCBI RefSeq mouse protein database in a target decoy fashion using X! Tandem Sledgehammer (2013.09.01.1). Search parameters used were: fixed modification (carboamidomethylation of cysteine; +57 Da), variable modifications (oxidation of methionine; +16 Da) and N-terminal acetylation; +42 Da), three missed tryptic cleavages, 20 ppm peptide mass tolerance and 0.6 Da fragment ion mass tolerance. The false discovery rate was <0.5%. Proteins were quantified using the Normalised Spectral Abundance Factor method[50]. Differentially expressed proteins between the $n = 8$ independent $Hdac3^{IKO}$ and $Hdac3^{WT}$ samples were identified using an unpaired Student's $t$ test, with $P < 0.05$ considered statistically significant.

**Lipidomic analysis.** Enterocytes were isolated by incubation of the entire small intestine in 15 mM EDTA for 30 min and pellets frozen. Samples in 300 µL of chilled PBS were homogenised using a Pro200 homogenizer (Pro Scientific), and 30 µg of protein used for lipid extraction as previously described[51]. Briefly, samples were lyophilised and reconstituted in 10 µl of water with 10 µl of internal standard mix (ISTD) added to each sample. Lipids were extracted by adding 200 µl of chloroform/methanol (2:1) and sonication for 30 min. The supernatant was transferred to a 96-well plate and dried under vacuum in a SpeedVac Concentrator (Savant). Samples were then reconstituted with 50 µl water saturated butanol and 50 µl methanol with 10 mM ammonium formate, and analysed by LC ESI-MS/MS using an Agilent 1200 LC system and an ABSciex 4000 Qtrap mass spectrometer. Chromatographic separation was performed using an Agilent Eclipse Plus C18 RRHD 2.1 × 100 mm 1.8-µm column. Mobile phases A and B consisted of water: acetonitrile:isopropanol, 50:30:20 and 1:9:90, respectively, and both contained 10 mM ammonium formate. The flow rate was 400 µl/min with a gradient of 10% B to 55% B over 3 min, increased to 70% B over 8 min, increased to 87% B over 0.1 min, increased to 92% B over 5.2 min, increased to 100% B over 0.1 min and held at 100% B for 0.3 min. The solvent was then decreased to 10% B over 0.1 min, and held at 10% B for 4.2 min until next injection at 21 min. The first 1.5 min and final 0.9 min of each analytical run were diverted to waste. The injection volume was 5 µl, and columns were heated to 60 °C for the run. The autosampler was maintained at 25 °C. The mass spectrometer used was an ABSciex Qtrap 4000 coupled to an Agilent 1200 HPLC system. For the run, all MRMs were monitored in positive mode, and the ESI-spray voltage was 5000 V and the temperature set to 350 °C. Lipids were measured using scheduled multiple reaction monitoring (sMRM), where data were collected over a 60 s retention time window. The results from the chromatographic data were analysed using MultiQuant 2.1.1, where relative lipid abundances were calculated by relating each area under the chromatogram for each lipid species to the corresponding internal standard. Lipids were separated and annotated first on MRM (precursor and product fragmentation pair) and then on retention time if there was any ambiguity.

The amount of internal standard added per sample is listed in the source data file (4E.1)[51].

**Assessment of fatty acid oxidation ex vivo.** Fatty acid oxidation was determined in the duodenum, liver and soleus muscle from $Hdac3^{WT}$ and $Hdac3^{IKO}$ mice. For the liver and muscle, tissue was incubated for 2 h with 1-$^{14}$C-Oleic acid (0.25 µCi/ml), oleic acid (0.5 mM) and BSA (2%) in low glucose DMEM media. For the intestine, 1.5 -cm sections of the duodenum or freshly isolated enterocytes from the entire small intestine were incubated for 30 min using the same medium. At the completion of the incubation period, the medium was collected, and released $^{14}$CO$_2$ measured in a scintillation counter (LS-6500 multipurpose scintillation counter; Beckman Coulter). Radioactive acid-soluble metabolites (ASM) in the medium and remaining in the cells were quantified separately and values pooled.

**Catalase and ACOX enzymatic activity assays.** For assessment of catalase and peroxisomal acyl-CoA oxidase (ACOX) activity, isolated enterocytes were homogenised in 50 mmol/L Tris-HCl, 1 mmol/L EDTA and 0.1% (v/v) Triton-X, pH 7.4. The ACOX enzyme assay is based on the production of hydrogen peroxide (H$_2$O$_2$) by ACOX using palmitoyl-CoA as a substrate, and the H$_2$O$_2$-dependent oxidation of H2-DCF (dichlorofluorescein; Thermo Fisher, VIC, Australia) by horseradish peroxidase (HRP), which can be assessed spectrophotometrically at 502 nm[52]. Briefly, a reaction buffer was prepared containing 25 mmol/L KH$_2$PO$_4$ (pH 7.4), 40 mmol/L aminotriazole, 0.08 mg/mL HRP and 0.05 mmol/L H$_2$-DCF. In all, 20 µL of cell

homogenate was mixed with 230 μL reaction buffer at 37 °C, and the enzymatic reaction was initiated through addition of 50 μL 0.3 mmol/L palmitoyl-CoA in 25 mmol/L $KH_2PO_4$. Change in absorbance was followed at 502 nm for 2–3 min, and ACOX activity in μmol/min/mg protein was calculated using the extinction coefficient for DCF (91 μmol/ml/cm).

For assessment of catalase activity, we used the peroxidatic function of catalase. This assay is based on the reaction of catalase with methanol in the presence of $H_2O_2$, generating formaldehyde which can be measured spectrophotometrically with 4-amino-3-hydrazino-5-mercapto-1,2,4-triazole (Purpald; Sapphire Bioscience, NSW, Australia) as the chromogen[53]. Briefly, 20 μL of cell homogenate or catalase standard was mixed with 100 μL 100 mmol/L $KH_2PO_4$, pH 7.0, 30 μL methanol and 20 μL 35.2 mmol/L $H_2O_2$, and incubated on a rocking platform at room temperature for 20 min. The reaction was terminated by addition of 30 μL of 10 mol/L KOH, followed by addition of 30 μL of 50 mmol/L Purpald. Absorbance was assessed at 540 nm 10 min followed addition of Purpald.

**Dietary and drug intervention studies**. Mice were routinely maintained on a STD diet (9% fat, 8720610; Barastoc Stockfeeds, Victoria, Australia). For dietary intervention studies, mice were fed either a 60% total fat diet (SF02-006; Specialty Feeds, Perth, Western Australia, Australia) for 4 weeks, or a 23.50% total fat diet (SF04-001; Specialty Feeds, Perth, Western Australia, Australia) for 11 weeks. To assess the impact of PPARα-agonist or HDAC3-inhibitor treatment on gene expression in vivo, mice were fed a standard chow diet supplemented with 0.1% w/w of the PPARα agonist WY-14643 or 150 mg/kg of the HDAC3 inhibitor RGFP966 for 8 days.

**Triglyceride assays**. Triglycerides from intestinal epithelial cells, liver and stool were extracted by the method of Folch[54] and determined by a spectrophotometric assay based in a Tinder type reaction from Thermo Fisher (Infinity Triglyceride Liquid Stable Reagent).

**Generation of small intestinal organoids**. Small intestinal organoids were generated from 6–8-week-old WT mice as according to the method of Sato et al.[55] with slight modifications. Briefly, the small intestine was incubated in ice-cold PBS with 2 mM EDTA. Intestinal crypts were isolated by mechanical tissue disruption after several washes with ice-cold PBS. Between 300 and 500, isolated crypts were resuspended in 50 μL of RGF BME Type2 Path Clear® Cultrex (Invitro technologies Cat.3533-001-02) and seeded in 24-well plates with 100 μl of Advanced DMEM/F12 (Invitrogen Cat. 12634-028) supplemented with 0.1% BSA, 1% glutamax, 10 mM HEPES, 1% Pen/Strep, 1x B27 (Invitrogen Cat. 12587-010), 1x N2 (Invitrogen Cat. 17502-048), 0.05 ng/μl EGF (prepotech AF-100-15), 0.1 ng/μl Noggin (prepotech 250-38) and 0.5 ng/μl mR-Spondin 1 (prepotech 315-32). Organoids were sub cultured a ratio of 1:5, and maintained in 24-well plates at a density of 100–300 organoids per well by passaging twice a week. Once villi formation was evident, organoids (four wells per condition) were treated with drug or vehicle control for 6 h.

**Chromatin immunoprecipitation**. Intestinal epithelial cells were isolated with PBS-EDTA buffer from 1-month-old $Hdac3^{WT}$ and $Hdac3^{IKO}$ mice. Chromatin was cross-linked with 1% formaldehyde and sonicated to produce DNA fragments of 200 to 800 bp. For each IP, 4 μg of HDAC3 antibody (Ab7030, Abcam) was used per 1000 μg of protein. Following reversal of crosslinks, DNA was purified using the ChIP DNA Clean and Concentrator kit (Zymo Research, Irvine, CA). PPAR response elements (PPREs) were identified in gene promoters using the Dragon PPRE spotter tool (V2.0), using the default threshold of 50%. Primers used for quantitative PCR are listed in Supplementary Table 4.

**Reporting summary**. Further information on research design is available in the Nature Research Reporting Summary linked to this article.

## Data availability
The authors declare that the data supporting the findings of this study are available within the paper and its supplementary information files and in the source data file. Raw data files for microarray data are available at the Gene Expression Omnibus via accession code GSE138775. The mass spectrometry proteomics data have been deposited to the ProteomeXchange Consortium via the PRIDE partner repository with the data set identifier PXD015738.

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

## Acknowledgements

This project was supported by NHMRC project grant (GNT1107836), NHMRC Senior Research Fellowships to J.M.M. (GNT1046092) and M.J.W. (GNT1077703), and the Operational Infrastructure Support Program, Victorian Government, Australia.

## Author contributions

M.D-S., M.K.M., C.M.R., R.N., I.N., H.A., S.A-O., A.L., C.M.S., P.I., H.K., S.K., L.T., A.R., S.J.G., D.S.W., P.Y., K B-A., M.J.W. and J.M.M: conducted, analysed and interpreted experiments. M.D-S., M.K.M., C.M.R., R.N., S.M., Y.G., A.M.S., M.J.W. and J.M.M: conceived, designed, interpreted experiments and/or supervised parts of the study. M.D-S., M.K.M., Y.G., S.H., A.M.S., M.J.W. and J.M.M: contributed to the writing of the paper.

## Competing interests

The authors declare no competing interests.
