## [Peer Review File · Nature Communications]

Reviewers' comments:

Reviewer #1 (Remarks to the Author):

Davalos-Salas et al reported a set of data concerning the metabolic impacts of intestinal-specific deletion of histone deacetylase 3 (HDac3) on fat depots in mice. According to the authors the observed effects are due to an indirect increase in enterocyte fatty acid oxidation and therefore limit lipid uptake.

The manuscript is well written; and regarding novelty, targeting intestinal fatty acid oxidation via Hdac3 is not new, other authors have modulated intestinal fatty acid oxidation via sirt3 overexpression (Ramachandran et al, *Molecular metabolism*, 2017). After reading the manuscript some major points arose while reading the manuscript:

- 1) In the introduction section the authors state that deletion of HDac3 in the intestinal epithelium affected genes associated with antimicrobial defense, alteration of gut microflora and increased the sensibility of the small intestine to damage and inflammation. These aspects were not evaluated in this manuscript? So why to use the same approach and suggest that HDac3 deletion in enterocyte might be a potential target to obesity? thee
- 2) The authors did not report the data regarding the villin-Cre mice (in both inducible and non-inducible system), this group has to be included as control group together with the floxed mice. In some cases, the Cre-mice themselves exhibit a phenotype. The villin cre mice with inducible system have to be injected too with tamoxifen and see if they have a phenotype.
- 3) The IpGTT and ITT, the area under the curve have to be calculated,
- 4) In spite of having no differences in food intake and physical activity, the energy generated from the oxidation of fat has to go somewhere?
- 5) Since fatty acid oxidation was increased, did the authors check the ketogenic pathway? Please report the levels of circulating ketone bodies.
- 6) The authors did not refer to other papers published in the recent years regarding the induction of fatty acid oxidation in the small intestine and its effect of energy homeostasis. This manuscript has to include the data of these manuscript in the discussion (Ramachandran et al., *Molecular Metabolism*, 2017; Karimian et al, *PLoS One*. 2013; 8(9): e74869 ; Karimian et al., Schober et al, *J Lipid Res*. 2013 May; 54(5): 1369–1384..
- 7) When the calorimetric measures were performed, were the mice adapted first before recording the data? Please specify if the mice were group-housed or single housed? If they were group-housed and then moved to calorimetric measurements by being single housed this constitute a huge stress for them and require more days to adapt for single housing.
- 8) Fatty acid oxidation tests were performed in the whole intestinal sample, however, this tissue is very heterogeneous and the deletion of Hdac3 concerns only enterocytes, therefore the test has to be performed in isolated enterocytes.
- 9) Where the enterocytes were isolated in another set of experiments for gene expression, did the authors check the purity of these cells by using specific markers if there are residual smooth muscle cells?
- 10) The western blot presented, the protein were they from the whole intestinal tissue or from the isolated enterocytes?

Reviewer #2 (Remarks to the Author):

Dávalos-Salas et al. examined the function of HDAC3 in intestinal epithelial cells (IECs) using a conditional knockout (KO) mouse model. They found that IECs-specific deletion of HDAC3 dramatically reduced the body weight on both normal chow and high fat diet (HFD), which is associated with increased expression of genes involved in fatty acid oxidation (FAO) and a slightly higher FAO rate in IECs. Considering that many of these FAO genes are known PPARα target genes and can be activated by PPARα agonists, the authors concluded that the activated PPARα signaling accounted for the lower body weight in IECs-specific HDAC3-KO mice.

The study addresses an important question. The data is overall clean. The manuscript is well

written. However, the mechanistic study is descriptive and there are several major concerns about data interpretation.

(1) The change in FAO enzymes levels and FAO rate in IECs was small. The contribution of IECs FAO to the whole-body FAO is also small. Therefore, the FAO in IECs per se is unlikely the major cause of the dramatic body weight phenotype. Convincing evidence, including those from necessity or sufficiency tests, is lacking to fully establish the cause-effect relationship.

(2) The authors have not fully considered or tested alternative explanations about the cause of the body weight phenotype. For example, could malabsorption be a factor? A simple measurement of TG content in the feces is not sufficient to rule it out.

(3) The glucose, insulin, blood lipid, IECs lipid, and liver lipid phenotypes are all likely the results of the lower body weight.

(4) The overlapping of HDAC3-downstream genes with PPAR α -downstream genes is not sufficient to establish the role of PPAR α in HDAC3-mediated gene regulation. There are many nuclear receptors or transcription factors that could interact with HDAC3 and potentially contribute to HDAC3-mediated FAO gene expression in IECs.

Reviewer #3 (Remarks to the Author):

In this paper, the authors characterize the phenotype of mice harbouring a deletion of Hdac3 in intestinal epithelium. They show that this deletion results in reduced weight gain and lower adiposity, together with improvement of metabolic parameters under standard and high fat diets. They propose a HDAC3/PPAR α axis controlling intestinal fatty acid oxidation capacity leading to the reduction of lipids available for systemic uptake. The concept is highly interesting but additional experiments should be performed in order to prove that 1/ Hdac3 intestinal specific deletion triggers an increase of β -oxidation, resulting into the lean phenotype in mice. Moreover, 2/ the molecular mechanisms of action should be studied in more detail. Finally, 3/ the in vivo response of these mice to treatment with PPAR α agonists should be assessed..

Additional major concerns :

- 1/ The reduced body weight gain phenotype of Hdac3 intestinal KO mice (Hdac3IKO) is impressive. However, it arises in advanced age (become statistically significant in 35 week-old mice and older). This phenotype can be an aging or senescent effect. Moreover, the metabolic phenotyping of these mice was performed irrespectively of any standard in term of age, with huge age variations (1 to 12 month-old). It appears to this reviewer as mandatory to provide more rigorous investigations in terms of basic metabolic rate, energy expenditure, food intake, physical activity, and respiratory exchange ratio, all done in age-standard adult mice. Moreover, monitoring metabolic improvement in high fat diet-fed adult mice would also appear as a more conventional approach.

- 2/ Hdac3 lox/lox mice were used as control mice all over the manuscript, what appears an arbitrary selection. This genetic modification can explain the phenotype observed by itself. The authors should also include a comprehensive characterization of VillinCre mice, compared to the phenotype of Hdac3IKO mice. Fasting glucose, insulin, triglyceride and cholesterol plasma levels should be added.

- 3/ Regarding enteric lipid absorption, the authors conclude on a reduced triglyceride absorption due to higher intestinal fatty acid oxidation. Robust experimental approaches addressing this important question are missing. A direct measurement of intestinal absorptive capacity using lipid tracers appears instrumental to clarify this point. In general, the lipoprotein profiling in fasted and post-prandial states have not been done, nor the effect on the response to PPAR α agonists were studied.

- 4/ The authors suggest that the coordinate induction of mitochondrial and peroxisomal β oxidation genes in Hdac3IKO mice occurs by de-repression of the PPAR α /HDAC3 regulated program. In support to this molecular explanation, they provide mRNA levels obtained in enteroids

and in small intestine of WT mice after treatment with a PPAR α agonist or HDAC3 inhibitor, showing parallel transcriptional induction. They also provide the manuscript with ChIP data showing HDAC3 recruitment to putative PPREs in the promoter of a subset of regulated genes. Intestinal epithelial cells express other major metabolic transcription factors and nuclear receptors (HNF4, LXRs, FXR) whose activity/expression was HDAC3-sensitive as reported in other tissues. The authors should investigate more deeply the central role of PPAR α in this process, using HDAC3/PPAR α re-ChIP or experiments in PPAR α KO organoids.

Minor points:

- OGTT, IPGTT and oral lipid tolerance should include iAUC calculations.
- Lack of information in most of the figure legends and methods renders interpretation of the presented data difficult: specification of the used statistical test, diet, sex and age of animals should be carefully mentioned. Which segment of the small intestine was used for IHC?
- Previous reports from the same group describe absence of Hdac3 as a positive phenotype for the mice. However, Alenghat et al (2013) reported that low levels of Hdac3 in the small intestine are directly related to IBD in human patients and a pro-inflammatory state of the small intestine in Hdac3IKO. The authors should discuss how they reconcile previous reports and their own observations?

Reviewers' comments:

Reviewer #1 (Remarks to the Author):

Davalos-Salas et al reported a set of data concerning the metabolic impacts of intestinal-specific deletion of histone deacetylase 3 (HDac3) on fat depots in mice. According to the authors the observed effects are due to an indirect increase in enterocyte fatty acid oxidation and therefore limit lipid uptake. The manuscript is well written; and regarding novelty, targeting intestinal fatty acid oxidation via Hdac3 is not new, other authors have modulated intestinal fatty acid oxidation via sirt3 overexpression (Ramachandran et al, Molecular metabolism, 2017). After reading the manuscript some major points arose while reading the manuscript:

1) In the introduction section the authors state that deletion of HDac3 in the intestinal epithelium affected genes associated with antimicrobial defense, alteration of gut microflora and increased the sensibility of the small intestine to damage and inflammation. These aspects were not evaluated in this manuscript? So why to use the same approach and suggest that HDac3 deletion in enterocyte might be a potential target to obesity?

We initiated our study of intestinal *Hdac3* as a potential target in obesity based on the reports that HDAC3 regulates the expression of lipid metabolism genes in several tissues, including the liver, heart and in macrophages. Given the intestinal epithelium plays a fundamental role in the uptake and delivery of lipids to peripheral tissues, and consumes ~20% of all incoming energy, we postulated that deletion of HDAC3 in this tissue may impact on obesity.

Subsequent to initiation of our study and our observation of the reduced body weight phenotype, the study by Alenghat *et al* was published in *Nature*, which reported a number of separate phenotypes including reduced Paneth cell numbers, elevated crypt elongation, and increased colonic inflammation in 3 month old mice. At this point, we carefully examined all of these phenotypes in our colony. Surprisingly, we did not find any evidence of these changes in our colony.

Notably, the phenotypes reported in the Alenghat study were completely lost when their animals were housed under germ-free conditions, indicating the inflammatory phenotypes observed were entirely dependent on commensal bacteria. The difference in phenotype of conventionally-housed mice between the Alenghat study and our own is therefore likely due to differences in the commensal bacterial populations between animal facilities. To confirm this, we re-established our colony at a second facility in Melbourne, but again, failed to observe the phenotypes reported by Alenghat *et al.* We subsequently resumed our studies focused on the role of Hdac3 in regulating lipid metabolism in enterocytes by regulating a PPAR-dependent gene program, and present these data herein.

We also point out that during the review of our manuscript, the Alenghat lab has published a follow up study in *Gastroenterology* (Aug 2018), in which they also now report a reduced body weight phenotype in their colony. In contrast to our study, their study focuses on altered expression of genes that regulate metabolism in response to the microbiota (*Chka*, *Mttp*, *Apoa1* and *Pck1*), notably none of which are altered in our cohort.

The impact of Hdac3 in the intestinal epithelium is therefore clearly complex and context dependent, resulting in both direct effects on enterocytes, and indirect effects which are mediated either as a consequence of impacting the composition of the gut microbiome, or which manifest differently depending on the inherent composition of the microbiota. We have significantly expanded on the discussion to summarize these findings and bring together these multiple observations.

2) The authors did not report the data regarding the villin-Cre mice (in both inducible and non-inducible system), this group has to be included as control group together with the floxed mice. In some cases, the Cre-mice themselves exhibit a phenotype. The villin cre mice with inducible system have to be injected too with tamoxifen and see if they have a phenotype.

In the manuscript we report the same phenotype in both the *Villin^{Cre}-Hdac3^{Lox/Lox}* mice (no tamoxifen) and *Villin^{CreER}-Hdac3^{Lox/Lox}* mice treated with tamoxifen, providing definitive evidence that the phenotypes induced following *Hdac3* inactivation are independent of tamoxifen. In addition, the body weight phenotype in *Villin^{CreER}-Hdac3^{Lox/Lox}* tamoxifen-treated mice manifests 11 weeks after the last tamoxifen treatment, indicating effects are unlikely to be tamoxifen-mediated.

However, we acknowledge that this does not account for possible off targets of *Cre* overexpression. To address this, we have re-derived the *Villin^{Cre}* and the *Villin^{CreER}* strains and aged the mice to the time points (or beyond) at which *Hdac3^{KO}* mice show a significant body weight difference. This process has taken us ~10 months to complete.

In new supplementary data presented in **Supplementary Figure 3A and B**, we demonstrate that the body weights of both male and female *Villin^{Cre}* and *WT* littermate controls aged up to 12 weeks are identical.

Similarly, *Villin^{CreER}* mice and *WT* littermate controls were also injected with tamoxifen, exactly as performed for the *Hdac3* knockout mice experiments. As shown in **Supplementary Figure 3C and 3D**, the body weights of both male and female *Villin^{CreER}* mice are identical to *WT* mice, in mice aged up to 14 weeks post tamoxifen injection. Finally, as requested by Reviewer 3, we have also determined the fasting glucose, insulin, triglyceride and cholesterol plasma levels in *Villin^{CreER}* tamoxifen-treated mice and matched controls. No difference in any of these metabolic readouts was observed between *Cre*

expressing and *WT* mice (**Supplementary Figure 3E-I**). We can therefore now definitively conclude that the phenotypes observed in *Hdac3*^{IKO} mice are not *Cre*-mediated.

A comprehensive analysis of the metabolic effects of *Villin*^{Cre} mice has not been reported despite its widespread use (including the Alenghat studies in *Nature* and *Gastroenterology*). Our finding that there is no metabolic phenotype in this strain therefore strengthens not only our findings but several other published studies (eg Xie et al, *JBC* 2006, Kabir et al, *JBC* 2015, Wang et al, *Cell Metabolism* 2016).

3) *The IpGTT and ITT, the area under the curve have to be calculated*

Our data presentation of GTTs and ITTs are consistent with widely accepted practice in the field. As described by a key methods paper describing assessment of glucose metabolism in mice, “The standard presentation of results from GTTs is a description of blood glucose levels over time after the glucose administration. Generally, a time course of absolute glucose levels is presented.” (Standard operating procedures for describing and performing metabolic tests of glucose homeostasis in mice. NIH Mouse Metabolic Phenotyping Center Consortium. *Dis Model Mech.* 2010 Sep-Oct; 3(9-10): 525–534).

Analysis of the blood glucose levels by two-way ANOVA compares the ‘area under the curve’ by assessing the main effect of genotype. This is how we, and many world-leading labs, present data from glucose tolerance tests (Meex et al. *Cell Metabolism*, 2015, PMID: 26603189; Turpin et al, *Diabetologia*, 2011, PMID: 20842343; Hevener et al., *J Clin Invest*, 2007, PMID: 17525798; Perry et al. *Science*, 2015, PMID: 25721504).

4) *In spite of having no differences in food intake and physical activity, the energy generated from the oxidation of fat has to go somewhere?*

We have now calculated the energy expenditure of *Hdac3*^{IKO} and *WT* littermate controls from the CLAMS data, and demonstrate that *Hdac3*^{IKO} mice have significantly increased energy expenditure. These data are now presented in **Figure 2E**.

5) *Since fatty acid oxidation was increased, did the authors check the ketogenic pathway? Please report the levels of circulating ketone bodies.* Ketone body formation from acetyl-CoA only occurs under circumstances such as fasting, starvation or prolonged strenuous exercise, where oxaloacetate levels are low due to its removal from the mitochondrion for conversion into glucose. When oxaloacetate is unavailable, acetyl-CoA is diverted to the formation of acetoacetate and β -hydroxybutyrate. We don’t expect oxaloacetate levels to be low in *Hdac3*^{IKO} mice so did not expect to see increased ketone body formation. To confirm this, we have now measured β -hydroxybutyrate levels in enterocytes of *Hdac3*^{WT} and *Hdac3*^{IKO} mice. As expected, we do not see any difference in the level of this ketone body. As levels are not changed in enterocytes, we did not measure levels in the serum.

6) The authors did not refer to other papers published in the recent years regarding the induction of fatty acid oxidation in the small intestine and its effect of energy homeostasis. This manuscript has to include the data of these manuscript in the discussion (Ramachandran et al., *Molecular Metabolism*, 2017; Karimian et al, *PLoS One*. 2013; 8(9): e74869 ; Karimian et al., Schober et al, *J Lipid Res*. 2013 May; 54(5): 1369–1384..

We now cite these papers in a new paragraph on page 16 and 17 discussing these and other prior studies which have investigated the potential for modulating intestinal fatty acid oxidation as a means for preventing obesity.

7) When the calorimetric measures were performed, were the mice adapted first before recording the data? Please specify if the mice were group-housed or single housed? If they were group-housed and then moved to calorimetric measurements by being single housed this constitute a huge stress for them and require more days to adapt for single housing.

Yes, we are aware of the time required for mice to adapt to single housing. In all calorimetric studies animals were adapted for 2-days prior to commencing measurements. This information has been added to the methods.

8) Fatty acid oxidation tests were performed in the whole intestinal sample, however, this tissue is very heterogeneous and the deletion of *Hdac3* concerns only enterocytes, therefore the test has to be performed in isolated enterocytes.

We agree, and have now repeated these experiments in isolated enterocytes. As shown below, the difference in fatty acid oxidation is even more pronounced in isolated enterocytes. We have added these data to **Figure 4B** of the manuscript.

9) Where the enterocytes were isolated in another set of experiments for gene expression, did the authors check the purity of these cells by using specific markers if there are residual smooth muscle cells?

We have now determined the percentage of epithelial cells in these preps by EpCAM staining and FACS and found 98% of the viable cells to be EpCAM positive. We have also examined expression of several smooth muscle cell markers in the microarray data generated from these enterocytes. As shown below, with the exception of *Hexim1*, expression levels of all of these markers were extremely low, indicating minimal contamination of muscle cells. Furthermore, with the exception of *Hexim1*, no difference on expression of these markers was observed between *Hdac3*^{WT} and *Hdac3*^{KO} cell preparations.

Contamination of residual smooth muscle cells is therefore unlikely to have impacted on any of the analyses performed on isolated enterocytes.

mRNA expression of *Hdac3* and muscle specific markers in enterocytes isolated from *Hdac3*^{WT} and *Hdac3*^{KO} mice. Data are taken from the same Illumina microarrays used to identify increased expression of lipid oxidation genes in *Hdac3*^{KO} mice. [Caldesmon (*Cald1*), alpha smooth muscle actin (*Acta1*), Calponin 1 (*Cnn1*), VE-cadherin (*Cdh5*), *Hexim 1* and Transgelin (*Tagln*)].

10) The western blot presented, the protein were they from the whole intestinal tissue or from the isolated enterocytes?

Western blots performed were on isolated enterocytes, hence the lack of HDAC3 protein in the knockout samples. These blots provide further evidence of the purity of the enterocyte isolation procedure.

Reviewer #2 (Remarks to the Author):

Dávalos-Salas et al. examined the function of HDAC3 in intestinal epithelial cells (IECs) using a conditional knockout (KO) mouse model. They found that IECs-specific deletion of HDAC3 dramatically reduced the body weight on both normal chow and high fat diet (HFD), which is associated with increased expression of genes involved in fatty acid oxidation (FAO) and a slightly higher FAO rate in IECs. Considering that many of these FAO genes are known PPARα target genes and can be activated by PPARα agonists, the authors concluded that the activated PPARα signaling accounted for the lower body weight in IECs-specific HDAC3-KO mice.

The study addresses an important question. The data is overall clean. The manuscript is well written. However, the mechanistic study is descriptive and there are several major concerns about data interpretation.

(1) The change in FAO enzymes levels and FAO rate in IECs was small. The contribution of IECs FAO to the whole-body FAO is also small. Therefore, the FAO in IECs per se is unlikely the major cause of the dramatic body weight phenotype. Convincing evidence, including those from necessity or sufficiency tests, is lacking to fully establish the cause-effect relationship.

We emphasize that the difference in body weight of the mice takes approximately 3 months to become first evident, and 6 months to 2 years to fully manifest (Figure 1). Viewed in this context, the body weight phenotype is in fact relatively subtle, and therefore consistent with a change in intestinal fatty oxidation.

In terms of experiments to establish cause and effect, we directly demonstrate that the rate of FAO is increased in the intestinal epithelium of *Hdac3*^{KO} mice, the levels of fatty acids, particularly long chain

fatty acids are significantly reduced in intestinal epithelial cells from *Hdac3^{IKO}* mice, and demonstrate that the difference in body weight is significantly accelerated when mice are challenged with a high fat diet. In addition, we demonstrate that lipid oxidation was unchanged in the muscle or liver, excluding the contribution of those tissues to the whole-body phenotype. In new experiments, we now also demonstrate that the effects of Hdac3 deletion can be pharmacologically recapitulated by treatment of mice with the HDAC3-specific inhibitor RGFP-966. Finally, underscoring the proposed mechanism that *Hdac3* deletion protects against diet-induced obesity by invoking constitutive expression of a PPAR driven gene expression program, we now demonstrate that *Hdac3^{IKO}* mice are acutely sensitive to weight loss following feeding a PPAR α agonist (Figure 6C, D).

(2) The authors have not fully considered or tested alternative explanations about the cause of the body weight phenotype. For example, could malabsorption be a factor? A simple measurement of TG content in the feces is not sufficient to rule it out.

To further investigate whether malabsorption may be a factor, we have now performed bomb calorimetry to determine the energy density of faecal samples collected from *Hdac3^{WT}* and *Hdac3^{IKO}* mice. As shown below, there is no difference in the energy density of the stool from *Hdac3^{WT}* and *Hdac3^{IKO}* mice fed either a standard diet or a high fat diet, indicating the reduced body weight of *Hdac3^{IKO}* mice is not due to malabsorption. We have added these data to Figure 2D (STD diet) and Supplementary Figure 7B (HFD) of the revised manuscript.

Energy content of stool samples collected from *Hdac3^{WT}* and *Hdac3^{IKO}* mice assessed by bomb calorimetry.

(3) The glucose, insulin, blood lipid, IECs lipid, and liver lipid phenotypes are all likely the results of the lower body weight.

We agree that the reduced glucose, insulin and liver lipid phenotypes are likely the result of the lower body weight. As stated in the second paragraph of the results section, the purpose of these experiments was to demonstrate the improved metabolic phenotype of leaner *Hdac3^{IKO}* mice.

However, we disagree that the difference in IEC lipids is a result of lower body weight as these experiments were performed in mice that were 1 month old, prior to developing a difference in body weight. Rather, our data support that the reduction in IEC lipids is due to increased expression of lipid oxidation genes and consequent increase in lipid oxidation in this tissue, which is directly supported experimentally (Figure 4B).

(4) The overlapping of HDAC3-regulation. There are many nuclear receptors or transcription factors

that could interact with HDAC3 and potentially contribute to HDAC3-mediated FAO gene expression in IECs.

We agree, and to resolve this we have now performed additional experiments in which we have now compared the gene expression changes induced by agonists of 4 other nuclear receptors which are expressed in the intestine (PPAR γ , PPAR δ , FXR and LXR) in organoids derived from WT mice. Agonists of PPAR γ , FXR and LXR induced only minor changes in expression of β -oxidation genes (no changes induced >2-fold), therefore we can conclude that these nuclear receptors are not major contributors to this transcriptional program (**Supplementary Figure 8**). Comparatively, the PPAR δ agonist GW501516 induced >2-fold expression of 3/12 genes (**Supplementary Figure 8**). We have therefore changed our interpretation of these findings to be inclusive of PPAR δ .

Reviewer #3 (Remarks to the Author):

In this paper, the authors characterize the phenotype of mice harbouring a deletion of Hdac3 in intestinal epithelium. They show that this deletion results in reduced weight gain and lower adiposity, together with improvement of metabolic parameters under standard and high fat diets. They propose a HDAC3/PPAR α axis controlling intestinal fatty acid oxidation capacity leading to the reduction of lipids available for systemic uptake. The concept is highly interesting but additional experiments should be performed in order to prove that 1/ Hdac3 intestinal specific deletion triggers an increase of β -oxidation, resulting into the lean phenotype in mice. Moreover, 2/ the molecular mechanisms of action should be studied in more detail. Finally, 3/ the in vivo response of these mice to treatment with PPAR α agonists should be assessed.

1. In regards to performing additional experiments to demonstrate that intestine-specific deletion of *Hdac3* triggers an increase in β -oxidation, we have now repeated the experiments in isolated enterocytes and observed a significantly higher rate of lipid oxidation in enterocytes isolated from *Hdac3*^{KO} mice. We have also included *Cre* mice as controls to confirm the effect is due to deletion of *Hdac3* and not overexpression of CRE. Finally, we have now tested the effect of treating C57BL6 WT mice with a small molecule HDAC3-specific inhibitor (RGFP-966) and demonstrated that this drug can inhibit weight gain in mice fed either a standard or a high fat diet, providing pharmacological confirmation of the genetic findings.

2. In regards to studying the molecular mechanism in more detail, we have now compared the gene expression changes induced by *Hdac3* with those of a PPAR α agonist as well as agonists of PPAR γ , PPAR δ , FXR and LXR in small intestinal organoids from WT mice. These findings indicate that in addition to PPAR α , de-repression of PPAR δ may also contribute to the increased expression of β -oxidation genes following *Hdac3* deletion.

3. As suggested, we have now assessed the response of *Hdac3*^{WT} and *Hdac3*^{KO} mice to treatment with a PPAR α agonist (**Figure 6C, D**). Please see detailed response in point 3 below.

Additional major concerns:

- 1/ The reduced body weight gain phenotype of Hdac3 intestinal KO mice (Hdac3IKO) is impressive. However, it arises in advanced age (become statistically significant in 35 week-old mice and older). This phenotype can be an aging or senescent effect.

In all cases, the controls are age matched so this cannot be an aging or senescent effect. Also, please note that the statistical significance highlighted in the original Figure 1 was of P values <0.005 , giving the appearance that the body weight difference becomes significant at 35 weeks. In fact, the body weight phenotype becomes statistically significant ($P<0.05$) at 11 weeks in *Villin^{Cre}-Hdac3^{lox/lox}* (Figure 1A-B), at 11 weeks post tamoxifen induction in *Villin^{CreER}-Hdac3^{lox/lox}* (Supplementary Figure 2C) mice fed a standard diet, and at 6 weeks post-induction in mice fed a high fat diet (Figure 5F). We have now revised the figures to indicate this.

Moreover, the metabolic phenotyping of these mice was performed irrespectively of any standard in term of age, with huge age variations (1 to 12 month-old). It appears to this reviewer as mandatory to provide more rigorous investigations in terms of basic metabolic rate, energy expenditure, food intake, physical activity, and respiratory exchange ratio, all done in age-standard adult mice.

We agree and have now repeated several experiments (daily activity, food intake, fecal output and energy density) in mice that are 4-8 weeks old (Revised Figure 2), so that there is now greater consistency throughout the manuscript.

In other cases, metabolic profiling was performed at later times to enable specific conclusions to be drawn. For example, the CT/MRI imaging (Figure 1D-E) and GTT, ITT and lipid tolerance tests (Figure 1G-J) were performed in 6 month old mice once they had fully developed a major body weight difference, to illustrate the point that leaner *Hdac3^{KO}* mice are protected from the co-morbidities associate with diet-induced obesity.

Moreover, monitoring metabolic improvement in high fat diet-fed adult mice would also appear as a more conventional approach.

The primary phenotype we report (i.e. reduced body weight, increase in expression of lipid oxidation genes and proteins, and remodeling of the lipidome in enterocytes) are all observed in mice fed a standard diet, therefore for consistency, the metabolic profiling was performed under similar conditions.

- 2/ *Hdac3 lox/lox* mice were used as control mice all over the manuscript, what appears an arbitrary selection. This genetic modification can explain the phenotype observed by itself. The authors should also include a comprehensive characterization of *Villin^{Cre}* mice, compared to the phenotype of *Hdac3^{KO}* mice. Fasting glucose, insulin, triglyceride and cholesterol plasma levels should be added.

To address this we have re-derived the *Villin^{Cre}* and the *Villin^{CreER}* strains and aged the mice to 3 months. In new supplementary data presented in Supplementary Figure 3, we demonstrate that the body weights of *Villin^{Cre}* and the *Villin^{CreER}* and WT littermate controls, are identical. As suggested, we have also determined the glucose, insulin, triglyceride and cholesterol plasma levels in the plasma of *Villin^{Cre}* and *Villin^{CreER}* tamoxifen-treated mice and matched controls in the fasted state. No difference in any of these metabolic readouts were observed between *Cre* expressing and WT mice (Supplementary Figure 3E-I). We can therefore now definitively conclude that the phenotypes observed in *Hdac3^{KO}* mice are not *Cre*-mediated.

- 3/ Regarding enteric lipid absorption, the authors conclude on a reduced triglyceride absorption due to higher intestinal fatty acid oxidation. Robust experimental approaches addressing this important question are missing. A direct measurement of intestinal absorptive capacity using lipid tracers appears instrumental to clarify this point. In general, the lipoprotein profiling in fasted and post-prandial states have not been done, nor the effect on the response to PPAR α agonists were studied.

To determine whether *Hdac3*^{IKO} mice have a defect in enteric lipid absorption, we have now performed bomb calorimetry analyses to determine the caloric content of the stool produced from *Hdac3*^{WT} and *Hdac3*^{IKO} mice. These analyses revealed no difference in the caloric content of the faeces of *Hdac3*^{WT} and *Hdac3*^{IKO} mice (or the fecal output), indicating the difference in body weight is not due to a difference in enteric lipid absorption. These data have been added to **Figure 2D** and **Supplementary Figure 7B**.

As suggested, we have also now performed lipoprotein profiling in the fasted and post-prandial states. As shown below, while LDL cholesterol levels were modestly elevated in *Hdac3*^{IKO} mice in the fasted state ($P=0.08$), there were no differences in LDL or HDL cholesterol levels between *Hdac3*^{WT} and *Hdac3*^{IKO} mice in the fed state.

Finally, as suggested, we have now assessed the response of *Hdac3*^{IKO} and *Hdac3*^{WT} mice to PPAR α agonist (WY14643) treatment. Addition of WY14643 into the diet reduced weight gain in both *Hdac3*^{WT} and *Hdac3*^{IKO} mice. Remarkably, and consistent with our proposed mechanism of action, the reduction in body weight was significantly more pronounced in *Hdac3*^{IKO} mice (**new Figure 6C**). Notably, this was despite food consumption in these mice being elevated, which may reflect a compensatory hyperphagia in these mice. These findings support our model that deletion of the transcriptional repressor HDAC3 enables more robust activation of PPAR α by agonist treatment, greater lipid oxidation in the intestine, and reduced weight gain. We thank the Reviewer for suggesting this important experiment which significantly strengthens the conclusions of the paper.

- 4/ The authors suggest that the coordinate induction of mitochondrial and peroxisomal β oxidation genes in *Hdac3*IKO mice occurs by de-repression of the PPAR α /HDAC3 regulated program. In support to this molecular explanation, they provide mRNA levels obtained in enteroids and in small intestine of WT mice after treatment with a PPAR α agonist or HDAC3 inhibitor, showing parallel transcriptional induction. They also provide the manuscript with ChIP data showing HDAC3 recruitment to putative PPREs in the promoter of a subset of regulated genes. Intestinal epithelial cells express other major metabolic transcription factors and nuclear receptors (HNF4, LXRs, FXR) whose activity/expression was HDAC3-sensitive as reported in other tissues. The authors should investigate more deeply the central role of PPAR α in this process, using HDAC3/PPAR α re-ChIP or experiments in PPAR α KO organoids.

The mitochondrial and peroxisomal lipid oxidation genes we focused on were those previously reported to be induced by a PPAR α agonist in WT mice but not in PPAR α knockout mice, and are therefore *bona fide* PPAR α target genes (Bunger *et al*, *Physiol Genomics* 2007). Nevertheless, to further investigate the central role of PPAR α in mediating the transcriptional response, we have now compared the gene expression changes induced by 4 other nuclear receptor agonists, PPAR γ , PPAR δ , FXR and LXR, in mouse

small intestinal organoids. While agonists of LXR, FXR, and PPAR γ had minimal effects on mitochondrial and peroxisomal β -oxidation genes, the PPAR δ agonist GW50156 induced 3/12 of the β -oxidation genes by 2-fold or greater. Based on these findings, we have altered our conclusions to be inclusive of PPAR δ .

With regards to re-ChIP experiments, we have attempted several times to perform the suggested HDAC3/PPAR α re-ChIP experiments in organoids, without success. Unfortunately, despite testing multiple PPAR α antibodies, we have not been able to successfully identify one which can immunoprecipitate PPAR α in mouse intestinal epithelial lysates. The limitation of these reagents is well established in the field. Instead, as summarized above, we have now also demonstrated that *Hdac3*^{IKO} mice have a significantly reduced weight gain when treated with a PPAR α agonist in the diet, consistent with the central role of PPAR nuclear receptors in driving the lipid oxidation phenotype in *Hdac3*^{IKO} mice.

Minor points:

- OGTT, IPGTT and oral lipid tolerance should include iAUC calculations.

Please see our response to Reviewer 1 above.

- Lack of information in most of the figure legends and methods renders interpretation of the presented data difficult: specification of the used statistical test, diet, sex and age of animals should be carefully mentioned.

We have now revised all the figure legends to include this information.

Which segment of the small intestine was used for IHC?

IHC stains were performed on sections of duodenum or the border of the duodenum and jejunum. We have added this information to the legend of **Supplementary Figure 4**.

- Previous reports from the same group describe absence of *Hdac3* as a positive phenotype for the mice. However, Alenghat et al (2013) reported that low levels of *Hdac3* in the small intestine are directly related to IBD in human patients and a pro-inflammatory state of the small intestine in *Hdac3*IKO. The authors should discuss how they reconcile previous reports and their own observations?

We agree that this is important issue to discuss. As outlined in detail in our response to Reviewer 1, despite extensive investigation, we have not observed evidence of an inflammatory phenotype in our cohort. Notably, the inflammatory phenotypes reported in the Alenghat study are completely lost when their animals are housed under germ-free conditions, and the conclusion of their study is that these phenotypes are dependent on commensal bacteria. The lack of an inflammatory phenotype in our cohort is therefore likely due to differences in the commensal bacterial populations of the 2 animal facilities.

Comparatively, while we did not observe any evidence of inflammation in our cohort, we observed a difference in body weight and a parallel induction of lipid oxidation genes, which became the focus of our study. Importantly, interrogation of the supplementary gene expression data from the Alenghat study revealed that the increased expression of PPAR targets/lipid oxidation genes we observe in *Hdac3*^{IKO} mice, is also evident in the Alenghat cohort, and importantly in both conventionally housed and germ-free mice, indicating this is an inherent phenotype of *Hdac3*^{IKO} mice. These points have now been added to the discussion.

We also wish to point out that in the August issue of *Gastroenterology*, the Alenghat lab published a follow up study (Whitt J *et al*, *Gastroenterology* Aug 2018) which supports some of our findings, particularly the reduced body-weight phenotype of *Hdac3*^{KO} mice. However, in contrast to our study, their study focuses on altered expression of genes which regulate metabolism in response to the microbiota (*Chka*, *Mttp*, *Apoa1* and *Pck1*).

The impact of *Hdac3* deletion in the intestinal epithelium is therefore clearly complex and context dependent, resulting in both direct effects on enterocytes, and effects which are mediated either as a consequence of impacting the composition of the gut microbiome, or which manifest differently depending on the inherent composition of the microbiota. We have significantly expanded on the discussion (**page 18**) to summarize these findings and bring together these multiple observations.

Reviewers' comments:

Reviewer #1 (Remarks to the Author):

The authors responded to most of the my concerns

However, there is still something missing which is the possible mechanistic by which the Hdac3 null mice are protected against DIO. The authors focus on the increased rate of enterocytes fatty acid oxidation that allows reducing dietary fat available for other organs. But a recent paper where fatty acid oxidation is increased specifically in the enterocytes in mice did not show any effect on body weight but rather glucose homeostasis when these mice are either fed low- or high-fat diet (Ramachandran et al. Scientific Reports, volume 8, N°: 10818, 2018. DOI: 10.1038/s41598-018-29139-6). This paper was not cited and it has to be put in the discussion and explain why when we increase specifically enterocyte fatty acid oxidation we do not observe any effect on body weight. The authors should include other possible mechanisms and other metabolic pathways that are controlled by Hdac3.

Reviewer #2 (Remarks to the Author):

The authors constructed an IEC-specific KO mouse model and observed a body weight (BW) phenotype. They performed RNA-seq and metabolomics analysis but were not able to connect the dots and form a solid explanation for the phenotype. As a result, the revised manuscript remained descriptive.

The slightly increased fatty acid oxidation (FAO) rate or PPARa activation in the endothelial cells is unlikely the reason for the BW phenotype for three reasons. (1) Increased PPARa activation would lead to higher, not lower, lipid content in the cell. This has been demonstrated in many previous studies with PPARa gain-of-function. (2) The contribution of IEC to whole-body FAO or energy expenditure is negligible. (3) There is no data on a rescue of FAO or BW phenotype with experimental manipulation of PPARa in the current study.

The experiment with the HDAC inhibitor is not relevant because it affects multiple tissues and is unlikely specific to HDAC3. The data with PPARg, FXR, and LXR agonists were over-interpreted because removal of a co-corepressor is not equivalent to adding an agonist. The experiment with WY14643 merely suggests that HDAC3 deficiency potentiates WY-mediated weight loss. It does not necessarily suggest that the BW phenotype in KO mice in the absence of WY is due to PPARa activation.

Reviewer #3 (Remarks to the Author):

The authors provide a revised version of the manuscript with additional data, demonstrating that intestinal HDAC3 invalidation and pharmacological inhibition, lead to an induction of fatty acid oxidation in enterocytes and also renders animals resistant to high fat diet-induced obesity. Especially, they now provide experimental data showing increased fatty acid oxidation capacity in intestinal epithelial cells of HDAC3^{iko} mice (figure 4B), and absence of modification in physical activity, food intake and no lipid malabsorption disorders in HDAC3^{iko} mice compared to controls (figure 2). The characterization of villin-cre mice is also provided, contributing to a more consistent overall phenotyping of HDAC3^{iko} mice. The approach is now much more rigorous regarding the animal group size and age variations, even though caution should be taken when interpreting data obtained from very young animals (4-6 wk old) (fig 2, 4, 8), which could lead to a bias.

However, there are still some key experiments missing, to support the hypothesis raised by the authors.

Major concerns:

1/ The authors propose that this impressive phenotype (lean phenotype under normal chow diet,

resistance to diet-induced obesity) in HDAC3^{iko} mice mostly results from an increase in fatty acid oxidation in the intestine, thus restricting the amount of lipids for triglyceride-rich lipoprotein (TRL) assembly, and decreasing consequently the dietary-derived lipid transfer to the plasma. It appears (fig 1I and fig 5D) that HDAC3^{iko} mice display a fasting hypotriglyceridemia, compared to control mice (whatever the diet), suggesting that HDAC3^{iko} intestines would not be the sole/major tissue with lipid metabolism alterations (since the intestine itself only marginally contributes to the control of fasting triglyceridemia, which is mainly dependent on liver VLDL output). Even given the calorimetric analyses of the faeces between Hdac3^{WT} and Hdac3^{IKO} (new figure 2D), it is not clear what the role of the intestine is as the principal organ involved in the chow diet phenotype. In this confusing situation, a lipid tolerance test combining the use of a lipid tracer and lipolysis inhibitor-treated mice is therefore mandatory to clarify whether or not, HDAC3^{iko} mice really display a decrease in intestinal TRL production capacity following an oral lipid challenge.

2/ The authors suggest that the induction of beta oxidation genes in Hdac3^{IKO} occurs by derepression of the intestinal PPAR α regulated program, but strong experimental evidences are still required to support this hypothesis. The authors added in vivo data with the PPAR α agonist Wy14643 and report that Wy14643 treatment induces a stabilization (control mice) or a decrease (HDAC3^{iko} mice) in body weight gain (fig 6C). What is the effect of Wy14643-treatment on beta-oxidation gene expression levels (mRNA levels) in enterocytes of HDAC3^{iko} mice, compared to control mice (fig 6A&B)? It appears essential to investigate in more detail the central role of PPAR α in this process. What is the effect of the HDAC3 inhibitor RGFP 966 (used in figure 8) in PPAR α knock out mice or intestinal organoids, in terms of beta oxidation gene expression levels and/or body weight gain ?

Minor points:

- OGTT, IPGTT and oral lipid tolerance should include iAUC calculations.
- Liver steatosis in 12 month Hdac3^{WT} mice under chow diet is surprising.
- Figures containing organoid-derived data : please specify in the legend the number of individual organoid preparations used in the 2 independent experiments"
- Information/detail inconsistency in most of the figure legends and methods still renders interpretation of the presented data difficult: specification of the used statistical test, diet, sex and age of animals should be carefully mentioned (cf list below) : Figure 1A-B: specify statistical test and diet used; Figure 1I and 2E : specify * = pvalue; Figure 3F-G: specify statistical test and indicate the LS-MS/MS method; •Figure 4A-E-H : specify sex; Figure 4F: specify statistical test; Figure 5B-C: specify sex; Figure 8: sex and statistical test; Supp Figure A: sex and statistical test and number and age of mice. Supp Figure 7A: statistical test. Supp figure 8: replicates. Supp figure 9: Add total legend information from the original paper
- The authors should cite papers studying the regulation of lipoprotein metabolism in the intestine and interpret their findings in light of these previous reports (eg gene expression changes in different apolipoprotein genes).

Reviewer #1 (Remarks to the Author):

The authors responded to most of the my concerns. However, there is still something missing which is the possible mechanistic by which the Hdac3 null mice are protected against DIO. The authors focus on the increased rate of enterocytes fatty acid oxidation that allows reducing dietary fat available for other organs. But a recent paper where fatty acid oxidation is increased specifically in the enterocytes in mice did not show any effect on body weight but rather glucose homeostasis when these mice are either fed low- or high-fat diet (Ramachandran et al. Scientific Reports, volume 8, N°: 10818, 2018. DOI: 10.1038/s41598-018-29139-6). This paper was not cited and it has to be put in the discussion and explain why when we increase specifically enterocyte fatty acid oxidation we do not observe any effect on body weight. The authors should include other possible mechanisms and other metabolic pathways that are controlled by Hdac3.

Response: All three Reviewer's express the need for us to consider and discuss the possibility that in addition to increasing the rate of lipid oxidation in enterocytes, intestinal-specific *Hdac3* deletion may be altering body weight by multiple mechanisms. We agree with the importance of this, and have undertaken an extensive re-write of the manuscript, particularly the discussion, to acknowledge and discuss alternate mechanisms (Page 16, track changes version of the manuscript provided).

We also now cite the suggested reference in the discussion (Ramachandran et al, 2018). We note that one point of difference between *Hdac3* deletion in the current study and mutant *Cpt1a* overexpression performed by Ramachandran et al, is our finding that *Hdac3* deletion impacts expression of both mitochondrial and peroxisomal lipid oxidation genes, which may contribute to the different phenotypes.

Reviewer #2 (Remarks to the Author):

The authors constructed an IEC-specific KO mouse model and observed a body weight (BW) phenotype. They performed RNA-seq and metabolomics analysis but were not able to connect the dots and form a solid explanation for the phenotype. As a result, the revised manuscript remained descriptive.

The slightly increased fatty acid oxidation (FAO) rate or PPAR α activation in the endothelial (sic epithelial) cells is unlikely the reason for the BW phenotype for three reasons. (1) Increased PPAR α activation would lead to higher, not lower, lipid content in the cell. This has been demonstrated in many previous studies with PPAR α gain-of-function. (2) The contribution of IEC to whole-body FAO or energy expenditure is negligible. (3) There is no data on a rescue of FAO or BW phenotype with experimental manipulation of PPAR α in the current study.

Response: Given the complexity and extent of effects induced by intestinal *Hdac3* deletion, we agree that directly linking the increase in enterocyte oxidation with the body weight changes is not technically feasible. Therefore, beginning with the title, we have revised the manuscript throughout to acknowledge this, and not to directly link the two phenotypes (please see track changes version).

Clarification of specific points:

(1) Increased PPAR α activation would lead to higher, not lower, lipid content in the cell. This has been demonstrated in many previous studies with PPAR α gain-of-function. While cardiac-specific transgenic overexpression of PPAR α has been shown to increase lipid levels in the myocardium (Finck et al, JCI 2002, Fink et al, PNAS 2003), multiple other studies have shown an opposite effect. Specifically, PPAR α agonists have been shown to reduce triglyceride levels in intestinal epithelial cells (Kimura et al. 2011; Kimura et al. 2013), hepatocytes (Veiga et al. 2017; Kim et al. 2018; Qin, Yin, and Huang 2016; Barbosa-

da-Silva et al. 2015) and macrophages (Ye et al. 2019). The reason for this disparity is unclear, but may be related to the different ways in which PPAR α is activated (overexpression versus ligand-mediated activation), tissue-specific differences, or the extent or duration of PPAR α activation. PPAR α agonists are also used in clinical practice due to their ability to reduce plasma TG levels (Han et al. 2017). There is therefore an extensive body of literature which supports our model that activation of PPAR enhances β -oxidation and reduces TG levels.

(2) The contribution of IEC to whole-body FAO or energy expenditure is negligible. Unfortunately no references are provided to guide our response to this statement. Nevertheless, our understanding is that the relative contribution of intestinal epithelial cells (IECs) to whole body FAO or energy expenditure is not well understood, as several studies which have quantified resting energy expenditure in different organs have omitted analysis of the GI tract due to technical limitations involved in studying this tissue (Gallagher et al. 2006). As cited in the paper, the intestinal epithelium has been reported to consume ~20% of all incoming energy to offset the energy required for the uptake of nutrients including lipids, and their packaging into chylomicrons for delivery to peripheral tissues (Cant, McBride, and Croom 1996). Furthermore, the two studies which we have found that have investigated energy expenditure in the gastrointestinal tract, estimate that 10-15% of total energy expenditure in humans occurs in the GI tract (Rolfe and Brown 1997; Frayn, Humphreys, and Coppack 1995). We therefore don't think it is accurate to completely dismiss the contribution of the intestinal epithelium to whole body FAO or energy expenditure.

(3) There is no data on a rescue of FAO or BW phenotype with experimental manipulation of PPAR α in the current study. We did initially attempt to perform rescue experiments in organoids using an antagonist of PPAR α which has been described in the literature (GW6471). However, disappointingly, the antagonist failed to inhibit the effects of PPAR α in this system, which prevented our ability to confidently perform further experiments with this compound. Furthermore, since we confirmed the Reviewer's suggestions in the first review that the transcriptional reprogramming in *Hdac3*^{IKO} mice is likely driven by multiple nuclear receptors, a proper rescue experiment would require simultaneous intestinal-specific inhibition/deletion of multiple nuclear receptors, which is not technically feasible.

The experiment with the HDAC inhibitor is not relevant because it affects multiple tissues and is unlikely specific to HDAC3.

We agree with the limitations of this experiment pointed out by the Reviewer and have removed these data from the manuscript.

The data with PPAR γ , FXR, and LXR agonists were over-interpreted because removal of a co-corepressor is not equivalent to adding an agonist.

The results obtained from treating organoids with agonists of PPAR γ , FXR and LXR are interpreted the same as for agonists of PPAR α and PPAR δ . With the PPAR γ , FXR and LXR agonists, we know the treatments have worked, as at least one gene in the panel was induced in each case, which alleviates this possible concern (**Supplementary Figure 8**). Nevertheless, we have re-written this section (page 13) and tempered the wording of the interpretation of these experiments.

On a separate note, in reviewing these experiments, we noted that for PPAR α , we had originally treated organoids with 5 μ M WY14643, whereas all other agonists were used at 1 μ M. We have now repeated the experiments with WY14643 at the 1 μ M dose to be in line with the other agonists (**Supplementary Figure 8**). 1 less gene was induced at the lower dose, otherwise the results obtained are the same.

The experiment with WY14643 merely suggests that HDAC3 deficiency potentiates WY-mediated weight loss. It does not necessarily suggest that the BW phenotype in KO mice in the absence of WY is due to PPAR α activation.

We agree that these data do not explain the body weight phenotype of *Hdac3*^{KO} mice in the absence of WY14643 and do not make this claim. As suggested by Reviewer 3, we have now examined induction of lipid oxidation genes in enterocytes from *Hdac3*^{KO} and *Hdac3*^{WT} mice treated with WY14643. These data clearly show that induction of genes, particularly those involved in peroxisomal lipid oxidation, is significantly more pronounced in enterocytes in *Hdac3*^{KO} mice. These findings confirm our model that deletion of *Hdac3* results in de-repression of lipid oxidation genes, and that a subset of these genes are PPAR α targets. We have limited our interpretation of these findings to that conclusion.

In summary, we have extensively revised the interpretations of the paper to focus predominantly on the role of HDAC3 in regulating lipid oxidation genes, and to not over-interpret the link between increased lipid oxidation in enterocytes and the reduction in body weight.

Reviewer #3 (Remarks to the Author):

The authors provide a revised version of the manuscript with additional data, demonstrating that intestinal HDAC3 invalidation and pharmacological inhibition, lead to an induction of fatty acid oxidation in enterocytes and also renders animals resistant to high fat diet-induced obesity. Especially, they now provide experimental data showing increased fatty acid oxidation capacity in intestinal epithelial cells of HDAC3iko mice (figure 4B), and absence of modification in physical activity, food intake and no lipid malabsorption disorders in HDAC3iko mice compared to controls (figure 2). The characterization of villin-cre mice is also provided, contributing to a more consistent overall phenotyping of HDAC3iko mice. The approach is now much more rigorous regarding the animal group size and age variations, even though caution should be taken when interpreting data obtained from very young animals (4-6 wk old) (fig 2, 4, 8), which could lead to a bias.

However, there are still some key experiments missing, to support the hypothesis raised by the authors.

Major concerns:

*1/ The authors propose that this impressive phenotype (lean phenotype under normal chow diet, resistance to diet-induced obesity) in HDAC3iko mice mostly results from an increase in fatty acid oxidation in the intestine, thus restricting the amount of lipids for triglyceride-rich lipoprotein (TRL) assembly, and decreasing consequently the dietary-derived lipid transfer to the plasma. It appears (fig 11 and fig 5D) that HDAC3iko mice display a fasting hypotriglyceridemia, compared to control mice (whatever the diet), suggesting that HDAC3iko intestines would not be the sole/major tissue with lipid metabolism alterations (since the intestine itself only marginally contributes to the control of fasting triglyceridemia, which is mainly dependent on liver VLDL output). Even given the calorimetric analyses of the faeces between *Hdac3*^{WT} and *Hdac3*^{KO} (new figure 2D), it is not clear what the role of the intestine is as the principal organ involved in the chow diet phenotype. In this confusing situation, a lipid tolerance test combining the use of a lipid tracer and lipolysis inhibitor-treated mice is therefore mandatory to clarify whether or not, HDAC3iko mice really display a decrease in intestinal TRL production capacity following an oral lipid challenge.*

The triglyceride levels shown in Figure 11 and 5D were following a 4h fast. Our interpretation was that differences in enterocyte oxidation may still be impacting serum TG levels at this time point. Nevertheless, we acknowledge that definitive conclusion of this is not possible. To address this, we have

now examined serum TGs following an overnight fast. As shown below, serum Tg levels are not different between *Hdac3*^{WT} and *Hdac3*^{KO} mice, indicating these mice do not have an inherent difference in liver VLDL output.

2/ The authors suggest that the induction of beta oxidation genes in *Hdac3*KO occurs by derepression of the intestinal PPAR α regulated program, but strong experimental evidences are still required to support this hypothesis. The authors added *in vivo* data with the PPAR α agonist Wy14643 and report that Wy14643 treatment induces a stabilization (control mice) or a decrease (*HDAC3*iko mice) in body weight gain (fig 6C). What is the effect of Wy14643-treatment on beta-oxidation gene expression levels (mRNA levels) in enterocytes of *HDAC3*iko mice, compared to control mice (fig 6A&B)?

We have now examined induction of lipid oxidation genes (and proteins) in enterocytes from *Hdac3*^{KO} and *Hdac3*^{WT} mice treated with WY14643. The data (**new Figure 7**) clearly shows that induction of these genes, particularly those involved in peroxisomal lipid oxidation, is enhanced in enterocytes from *Hdac3*^{KO} compared to *Hdac3*^{WT} mice. These findings confirm our model that *Hdac3* deletion results in de-repression of PPAR α target genes.

It appears essential to investigate in more detail the central role of PPAR α in this process. What is the effect of the HDAC3 inhibitor RGFP 966 (used in figure 8) in PPAR α knockout mice or intestinal organoids, in terms of beta oxidation gene expression levels and/or body weight gain?

As per the the Reviewer's comments from the initial review of the paper, we examined and demonstrated that the effect of *Hdac3* deletion on expression of lipid oxidation genes is indeed mediated by de-repression of multiple nuclear receptors, including PPAR α and PPAR δ (Supplementary Figure 8). Therefore, in addition to being time prohibitive (~ 12 months to obtain and generate intestinal-specific PPAR α KO mice / organoids), studies in PPAR α IKO mice alone are unlikely to definitively confirm the *Hdac3*^{KO} phenotype. For example, intestinal-specific PPAR δ mice have recently been shown to be more prone to high fat diet-induced obesity (Doktorova et al. 2017), which is consistent with our proposed model. This study is now cited in the discussion (page 15) and we highlight the advantage of targeting HDAC3 compared to any specific PPAR as a potentially more effective means of manipulating lipid oxidation in the intestine.

Minor points:

- OGTT, IPGTT and oral lipid tolerance should include iAUC calculations.

These data have now been added to Figure 1. Panels G, H and I.

- Liver steatosis in 12 month *Hdac3*^{WT} mice under chow diet is surprising.

The livers from 12 month old *Hdac3*^{WT} mice were examined by a clinical pathologist in a blinded fashion who made this observations.

- Figures containing organoid-derived data: please specify in the legend the number of individual organoid preparations used in the 2 independent experiments”

This information has been added to the supplementary figure legend.

- Information/detail inconsistency in most of the figure legends and methods still renders interpretation of the presented data difficult: specification of the used statistical test, diet, sex and age of animals should be carefully mentioned (cf list below) : Figure 1A-B: specify statistical test and diet used; Figure 1I and 2E : specify * = pvalue; Figure 3F-G: specify statistical test and indicate the LS-MS/MS method;•Figure 4A-E-H : specify sex; Figure 4F: specify statistical test; Figure 5B-C: specify sex; Figure 8: sex and statistical test; Supp Figure A: sex and statistical test and number and age of mice. Supp Figure 7A: statistical test. Supp figure 8: replicates. Supp figure 9: Add total legend information from the original paper

All of these have points have been corrected, thank you.

- The authors should cite papers studying the regulation of lipoprotein metabolism in the intestine and interpret their findings in light of these previous reports (eg gene expression changes in different apolipoprotein genes).

Response: We have now examined expression of genes involved in lipoprotein metabolism (chylomicron assembly) in enterocytes (*Mttp*, *ApoA1*, *ApoA2*, *ApoA4*) for which microarray data was available, as part of examining whether intestinal fatty acid uptake, intracellular processing or secretion may contribute to the phenotype of the mice (**Supplementary Figure 7C**). No significant differences in expression of these genes were observed between *Hdac3*^{WT} and *Hdac3*^{KO} mice. A probe for *Apob* was not present on the microarray so we assessed expression by qPCR. As shown below, a modest (1.5-fold) induction was observed in *Hdac3*^{KO} mice. Overall, given apolipoprotein genes are not downregulated in *Hdac3*^{KO} mice, we can conclude that does not contribute to the phenotype of *Hdac3*^{KO} mice.

- Barbosa-da-Silva, S., V. Souza-Mello, D. C. Magliano, S. Marinho Tde, M. B. Aguila, and C. A. Mandarim-de-Lacerda. 2015. 'Singular effects of PPAR agonists on nonalcoholic fatty liver disease of diet-induced obese mice', *Life Sci*, 127: 73-81.
- Cant, J. P., B. W. McBride, and W. J. Croom, Jr. 1996. 'The regulation of intestinal metabolism and its impact on whole animal energetics', *J Anim Sci*, 74: 2541-53.
- Doktorova, M., I. Zwarts, T. V. Zutphen, T. H. V. Dijk, V. W. Bloks, L. Harkema, A. Bruin, M. Downes, R. M. Evans, H. J. Verkade, and J. W. Jonker. 2017. 'Intestinal PPARdelta protects against diet-induced obesity, insulin resistance and dyslipidemia', *Sci Rep*, 7: 846.
- Frayn, K. N., S. M. Humphreys, and S. W. Coppack. 1995. 'Fuel selection in white adipose tissue', *Proc Nutr Soc*, 54: 177-89.
- Gallagher, D., J. Albu, Q. He, S. Heshka, L. Buxt, N. Krasnow, and M. Elia. 2006. 'Small organs with a high metabolic rate explain lower resting energy expenditure in African American than in white adults', *Am J Clin Nutr*, 83: 1062-7.
- Han, L., W. J. Shen, S. Bittner, F. B. Kraemer, and S. Azhar. 2017. 'PPARs: regulators of metabolism and as therapeutic targets in cardiovascular disease. Part I: PPAR-alpha', *Future Cardiol*, 13: 259-78.
- Kim, M. J., C. H. Park, D. H. Kim, M. H. Park, K. C. Park, M. K. Hyun, A. K. Lee, H. R. Moon, and H. Y. Chung. 2018. 'Hepatoprotective Effects of MHY3200 on High-Fat, Diet-Induced, Non-Alcoholic Fatty Liver Disease in Rats', *Molecules*, 23.
- Kimura, R., N. Takahashi, S. Lin, T. Goto, K. Murota, R. Nakata, H. Inoue, and T. Kawada. 2013. 'DHA attenuates postprandial hyperlipidemia via activating PPARalpha in intestinal epithelial cells', *J Lipid Res*, 54: 3258-68.
- Kimura, R., N. Takahashi, K. Murota, Y. Yamada, S. Niiya, N. Kanzaki, Y. Murakami, T. Moriyama, T. Goto, and T. Kawada. 2011. 'Activation of peroxisome proliferator-activated receptor-alpha (PPARalpha) suppresses postprandial lipidemia through fatty acid oxidation in enterocytes', *Biochem Biophys Res Commun*, 410: 1-6.
- Qin, S., J. Yin, and K. Huang. 2016. 'Free Fatty Acids Increase Intracellular Lipid Accumulation and Oxidative Stress by Modulating PPARalpha and SREBP-1c in L-02 Cells', *Lipids*, 51: 797-805.
- Rolfe, D. F., and G. C. Brown. 1997. 'Cellular energy utilization and molecular origin of standard metabolic rate in mammals', *Physiol Rev*, 77: 731-58.
- Veiga, F. M. S., F. Graus-Nunes, T. L. Rachid, A. B. Barreto, C. A. Mandarim-de-Lacerda, and V. Souza-Mello. 2017. 'Anti-obesogenic effects of WY14643 (PPAR-alpha agonist): Hepatic mitochondrial enhancement and suppressed lipogenic pathway in diet-induced obese mice', *Biochimie*, 140: 106-16.

Ye, G., H. Gao, Z. Wang, Y. Lin, X. Liao, H. Zhang, Y. Chi, H. Zhu, and S. Dong. 2019. 'PPARalpha and PPARgamma activation attenuates total free fatty acid and triglyceride accumulation in macrophages via the inhibition of Fatp1 expression', *Cell Death Dis*, 10: 39.

REVIEWERS' COMMENTS:

Reviewer #1 (Remarks to the Author):

The authors responded to my concerns

Reviewer #2 (Remarks to the Author):

The authors have addressed my concerns

Reviewer #3 (Remarks to the Author):

Overall, the authors responded to most of our concerns, except for the analysis of intestinal-derived lipid production, thus leaving as an unresolved point, the mechanistic link between enterocyte lipidome remodelling and metabolic improvement. Even if alternate mechanisms may contribute to metabolic phenotype of HDAC3^{iko} mice, it appears of particular interest, in absence of experimental data, that the authors discuss how HDAC3 invalidation in the enterocyte could impact, either in a qualitative or in a quantitative manner, on lipid and lipoprotein secretion by the gut in the post-prandial period.

Overnight fasting triglyceridemia (data provided in the rebuttal) should be included as supplemental data.

Caution should still be taken when interpreting data obtained from very young animals (4-8 wk old), which could lead to a bias. Authors should clarify this point in the manuscript.

Reviewer #4 (Remarks to the Author):

The revised manuscript under review here; Deletion of intestinal Hdac3 remodels the lipidome of enterocytes and protects mice from 2 diet-induced obesity, by Mercedes Dávalos-Salas et al., is a thorough characterization of the role of HDAC3 on lipid metabolism, via regulation of lipid metabolic enzymes, in the intestinal epithelium.

This is a careful and broad analysis of the consequences of conditional deletion of HDAC3 from the intestinal epithelium, chief of which is the coordination of PPAR-regulated lipid catabolism.

This is a primarily descriptive study, though powerful in its scope. The authors recognize the limitations in drawing mechanistic interpretations from the perturbation of complex genetic networks of metabolic gene regulation with their associated primary and secondary effects observed at the tissue level. In this regard, the authors have been responsive to prior critique of this work.

Despite the overall descriptive nature of the work, significant insight into the underlying tissue homeostatic regulatory mechanisms, in particular with regard to tissue-specific regulation of lipid catabolic/anabolic balance by HDAC3 comprises very useful information.

Point by point response to Reviewer's comments:

We thank the Reviewer's for their review of our manuscript and their valuable feedback.

Reviewer #1 (Remarks to the Author):

The authors responded to my concerns.

Reviewer #2 (Remarks to the Author):

The authors have addressed my concerns.

Reviewer #3 (Remarks to the Author):

1. Overall, the authors responded to most of our concerns, except for the analysis of intestinal-derived lipid production, thus leaving as an unresolved point, the mechanistic link between enterocyte lipidome remodelling and metabolic improvement. Even if alternate mechanisms may contribute to metabolic phenotype of HDAC3^{KO} mice, it appears of particular interest, in absence of experimental data, that the authors discuss how HDAC3 invalidation in the enterocyte could impact, either in a qualitative or in a quantitative manner, on lipid and lipoprotein secretion by the gut in the post-prandial period.

We have added the following sentence to the discussion:

We also acknowledge that while expression of genes involved in lipid absorption and lipoprotein secretion were not altered in Hdac3^{KO} mice, additional studies are required to determine if Hdac3 deletion in enterocytes may contribute to the metabolic phenotype by impacting on lipid or lipoprotein secretion.

2. Overnight fasting triglyceridemia (data provided in the rebuttal) should be included as supplemental data.

We have added this data to Supplementary Figure 1. The following text has also been added to the "Intestinal deletion of Hdac3 reduces adiposity" section of the results:

Comparatively, serum Tg levels were not different between Hdac3^{WT} and Hdac3^{KO} mice fasted overnight indicating these mice do not have an inherent difference in liver VLDL output (Supplementary Figure 1D).

3. Caution should still be taken when interpreting data obtained from very young animals (4-8 wk old), which could lead to a bias. Authors should clarify this point in the manuscript.

We have added the following sentence to the methods section entitled "Generation of Hdac3^{KO} and Hdac3^{KO-IND} mice"

We note that as the majority of experiments were performed in 4-8 week old mice interpretation of findings are primarily applicable to this age group.

Reviewer #4 (Remarks to the Author):

The revised manuscript under review here; Deletion of intestinal Hdac3 remodels the lipidome of enterocytes and protects mice from diet-induced obesity, by Mercedes Dávalos-Salas et al., is a thorough characterization of the role of HDAC3 on lipid metabolism, via regulation of lipid metabolic enzymes, in the intestinal epithelium.

This is a careful and broad analysis of the consequences of conditional deletion of HDAC3 from the intestinal epithelium, chief of which is the coordination of PPAR-regulated lipid catabolism.

This is a primarily descriptive study, though powerful in its scope. The authors recognize the limitations in drawing mechanistic interpretations from the perturbation of complex genetic networks of metabolic gene regulation with their associated primary and secondary effects observed at the tissue level. In this regard, the authors have been responsive to prior critique of this work.

Despite the overall descriptive nature of the work, significant insight into the underlying tissue homeostatic regulatory mechanisms, in particular with regard to tissue-specific regulation of lipid catabolic/anabolic balance by HDAC3 comprises very useful information.

We thank the Reviewer for their evaluation of this study.